# Hyperglycaemia inhibits REG3A expression to exacerbate TLR3-mediated skin inflammation in diabetes

Yelin Wu[1,*], Yanchun Quan[1,*], Yuanqi Liu[1], Keiwei Liu[1], Hongquan Li[1], Ziwei Jiang[1], Tian Zhang[1], Hu Lei[1], Katherine A. Radek[2], Dongqing Li[1], Zhenhua Wang[1], Jilong Lu[1], Wang Wang[1], Shizhao Ji[3], Zhaofan Xia[3] & Yuping Lai[1]

Dysregulated inflammatory responses are known to impair wound healing in diabetes, but the underlying mechanisms are poorly understood. Here we show that the antimicrobial protein REG3A controls TLR3-mediated inflammation after skin injury. This control is mediated by REG3A-induced SHP-1 protein, and acts selectively on TLR3-activated JNK2. In diabetic mouse skin, hyperglycaemia inhibits the expression of IL-17-induced IL-33 via glucose glycation. The decrease in cutaneous IL-33 reduces REG3A expression in epidermal keratinocytes. The reduction in REG3A is associated with lower levels of SHP-1, which normally inhibits TLR3-induced JNK2 phosphorylation, thereby increasing inflammation in skin wounds. To our knowledge, these findings show for the first time that REG3A can modulate specific cutaneous inflammatory responses and that the decrease in cutaneous REG3A exacerbates inflammation in diabetic skin wounds.

[1] Shanghai Key Laboratory of Regulatory Biology, School of Life Sciences, East China Normal University, Shanghai 200241, China. [2] Department of Surgery, Burn and Shock Trauma Research Institute, Loyola University Chicago, Health Sciences Campus, Maywood, Illinois 60153, USA. [3] Burn Institute of Chinese PLA and Department of Burn Surgery, Changhai Hospital, Second Military Medical University, Shanghai 200433, China. * These authors contributed equally to this work. Correspondence and requests for materials should be addressed to Y.L. (email: yplai@bio.ecnu.edu.cn).

Diabetic patients exhibit delayed healing of acute wounds, which often develop into chronic ulcers in their feet and lower limbs. Multiple and complex pathophysiological factors contribute to this failure to heal[1–6]. Among these factors, excessive inflammatory cytokines lead to dysregulated leukocyte influx, thus resulting in sustained inflammatory responses and poor healing. Although diabetes is characterized by dysfunctional immune responses due to the metabolic impairment, the mechanisms by which these inflammatory responses contribute to impaired wound healing remains largely unknown.

To ensure an integrated progression of a distinct wound healing, several types of cells participate in various stages of the wound healing process. Keratinocytes are the predominant epithelial cell in skin and execute several complex processes that trigger inflammation as well as cell proliferation to maintain the structure and function of the epidermis[7]. After skin injury, the proliferation and migration of epidermal keratinocytes are crucial for wound re-epithelialization, but diminished keratinocyte proliferation and migration are observed in chronic non-healing wounds of diabetic patients[8]. Accumulating evidence demonstrates that these cellular deficits have been linked to increased production of inflammatory cytokines. Compared with wounds from normal patients, the production of pro-inflammatory cytokines, such as tumour necrosis factor (TNF)-α and interleukin (IL)-6, is dramatically increased to induce prolonged leukocyte infiltration in diabetic wound tissues[4,9,10], while the neutralization of TNF-α inhibits keratinocyte apoptosis to improve wound closure in diabetic wounds[11–14]. These observations suggest that the dynamic regulation of pro-inflammatory cytokine production might be an effective strategy for the management of wound repair in diabetes.

In skin injury, the expression of pro-inflammatory cytokines occurs in part through the activation of Toll-like receptors (TLRs)[15]. Overexpression or sustained activation of TLRs would lead to several inflammatory disorders including psoriasis and diabetic ulcers[15–21]. TLR3 is a key element in the initiation of inflammatory responses after skin injury[15], and enhanced expression of TLR3 has been observed in macrophages from non-obese diabetic mice[18]. In addition to TLR3, the aberrant activation of other TLRs, such as TLR2 and TLR4, in diabetes induces hyper production of pro-inflammatory cytokines[18,19], thus leading to tissue destruction[20–22]. These observations have raised the possibility that the control of TLR-induced inflammatory responses in diabetic wounds may improve wound repair.

To limit TLR-induced inflammation, multiple negative regulatory factors have been reported to regulate the transduction of TLR signalling including TNF receptor-associated factor 1 (TRAF1), Src homology region 2 domain-containing protein tyrosine phosphatase 1 (SHP-1), TNF-α-induced protein-3 (TNFAIP3), and sterile α and TIR motif-containing 1 protein (SARM1)[15,23]. Besides negative regulators, several antimicrobial peptides or proteins have shown anti-inflammatory effects[24–26]. For instance, the antimicrobial peptide, LL-37, inhibits hyaluronan-activated TLR4/CD44 signalling to decrease the expression of pro-inflammatory cytokines[24], and the antimicrobial protein, regenerating islet-derived protein 3A (REG3A), has been implicated to regulate uncontrolled inflammation by reducing pro-inflammatory cytokine production in ulcerative colitis[27]. However, whether REG3A enables to control TLR-mediated inflammatory responses in diabetic skin wounds remains completely unknown.

Given the importance of TLRs in inflammatory responses and the potential protective role of REG3A after tissue damage, we set out to investigate whether REG3A would be involved in homeostatic control of TLR3-mediated inflammation in skin wounds. Our findings uncover a previous unknown mechanism by which the antimicrobial protein REG3A suppresses skin inflammation after injury, and reveal that the decrease in cutaneous REG3A production amplifies inflammation in skin wounds of diabetic patients.

## Results

**REG3A defect amplifies inflammation in diabetic skin wounds.** Wound healing in diabetic patients is frequently impaired owing to hyper production of pro-inflammatory cytokines, and the antimicrobial protein REG3A has been implicated to regulate uncontrolled inflammatory responses in ulcerative colitis[27]. We thereby hypothesized that impaired wound healing in diabetes might correlate with an aberrant REG3A expression. To test this, we analysed REG3A expression on biopsies from the skin of five diabetic patients. Compared with normal (non-diabetic) patients with acute injury, the expression of REG3A mRNA was significantly decreased in acute wounds of diabetic patients (Fig. 1a). Consistent with decreased mRNA expression, REG3A protein expression was markedly decreased in epidermal keratinocytes around acute wounds from diabetic patients (Fig. 1b). To confirm that REG3A is related to impaired wound healing in diabetes, we evaluated a streptozotocin (STZ)-induced experimental type 1 diabetic (T1D) mouse model and found that both mRNA and protein of RegIIIγ, a mouse homologue of human REG3A, dramatically decreased in skin wounds of C57BL/6 T1D mice (Fig. 1c–e). RegIIIγ mRNA expression was also decreased in skin wounds of STZ-induced BALB/c T1D, genetically obese leptin-receptor-deficient ($Lepr^{db/db}$) and non-obese diabetic mice (Supplementary Fig. 1a–c). To further confirm that the decrease in epidermal RegIIIγ production might increase the expression of pro-inflammatory cytokines to impair wound healing in diabetes, RegIIIγ was intradermally injected into the dorsal skin of normal and T1D mice before wounding. The application of RegIIIγ significantly inhibited both mRNA and protein of TNF-α and IL-6 in the skin wounds of normal and T1D mice (Fig. 1f,g). This correlated with a decrease in inflammation and leukocytes infiltration (Supplementary Fig. 1d). Consistent with decreased inflammation, RegIIIγ significantly accelerated wound healing in T1D mice (Fig. 1h,i). Altogether, these data confirm that RegIIIγ is required for the control of inflammatory responses in skin wounds, and that defective REG3A/RegIIIγ expression leads to impaired wound healing in diabetes.

**Hyperglycaemia inhibits IL-33 to decrease REG3A production.** Our previous observation has demonstrated that IL-17 induces REG3A expression in keratinocytes[28]. We thereby first examined whether the production of IL-17 would be decreased similarly as RegIIIγ in the skin wounds of diabetic mice. Unexpectedly, IL-17 production was not decreased in the skin wounds of T1D mice compared with that in the skin wounds of normal mice (Fig. 2a and Supplementary Fig. 2a). We then screened other cytokines or TLR ligands that might influence REG3A expression in keratinocytes. Neonatal human epidermal keratinocytes (NHEKs) treated with a panel of cytokines or TLR ligands showed that recombinant human IL-33(rhIL-33) and rhIL-36γ, but not other cytokines or TLR ligands, induced REG3A expression in keratinocytes (Supplementary Fig. 2b). The induction of REG3A by rhIL-33 and rhIL-36γ was dose-dependent in both neonatal and adult human epidermal keratinocytes (Fig. 2b, Supplementary Fig. 2c–e). Therefore, we hypothesized that the reduction of REG3A might be due to decreased IL-33 or IL-36γ in diabetic skin

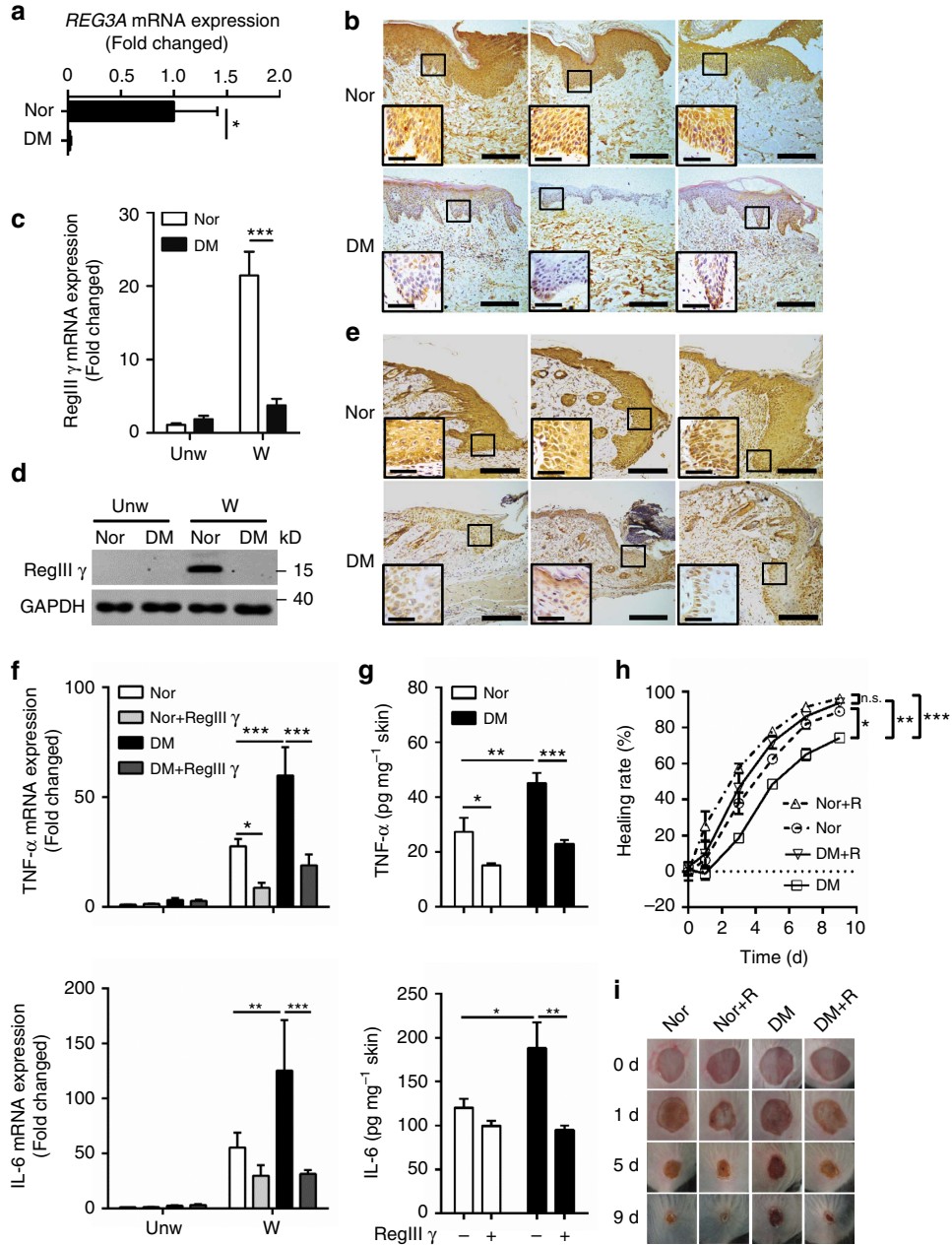

**Figure 1 | REG3A defect amplifies inflammation to impair wound healing in diabetes.** (**a**) Quantification of REG3A mRNA expression in the skin wounds of normal or diabetic patients with acute injury ($n=5$). (**b**) Immunohistochemical analysis of REG3A in the skin wounds from three normal or diabetic patients. The samples from normal patients were taken 1 day (two patients) or 2 days after acute injury, whereas the samples from diabetic patients were taken 5 or 9 or 14 days after acute injury. (**c**) The expression of RegIIIγ mRNA in skin wounds of normal and T1D mice 3 days after aseptic injury ($n=6$). (**d**) Immunoblot analysis of RegIIIγ in skin extracts taken from 2 mm surrounding the wound edges of normal and T1D mice. (**e**) Immunohistochemical analysis of RegIIIγ in day-3 skin wounds of normal and T1D mice. (**f**) Quantification of TNF-α and IL-6 mRNA expression in day-3 skin wounds from normal and T1D mice treated with or without 100 μg RegIIIγ ($n=8$). (**g**) Quantification of TNF-α and IL-6 production by ELISA in day-3 skin wounds of normal and T1D mice ($n=5$). (**h**) Wound healing of normal and T1D mice treated with ($n=7$) or without 100 μg RegIIIγ ($n=6$). (**i**) Photographs of healing wounds from normal and T1D mice treated as in **h**. All the mice used in the study without specific notes were C57BL/6 strain, and T1D mice were developed by STZ induction. Long scale bars represent 200 μm. Short scale bars represent 50 μm. Black rectangles designate region of ×400 magnification shown in inset of **b** and **e**. The abbreviations used here are normal (Nor), diabetes mellitus (DM), unwounded (unW), wounded (W) and RegIIIγ (R). *$P<0.05$, **$P<0.01$ and ***$P<0.001$. $P$ values were determined by two-tail $t$-tests (**a**) or two-way analysis of variance (ANOVA; **c,f–h**). Data are means ± s.e.m. and representative of two to three independent experiments.

wounds. The *in vivo* experiments showed that IL-33, but not IL-36γ, was decreased in a similar expression pattern of RegIIIγ in the skin wounds of T1D mice, as well as patients with diabetes (Fig. 2c–e and Supplementary Fig. 2a). Moreover, the intradermal administration of recombinant murine IL-33 (rmIL-33) into the dorsal skin of wild-type

(WT) T1D mice before wounding restored RegIIIγ production in skin wounds (Fig. 2f, Supplementary Fig. 2f) and significantly accelerated wound healing in T1D mice (Fig. 2g). These data demonstrate that the reduction of IL-33 leads to the decrease in cutaneous REG3A/RegIIIγ in diabetic skin wounds.

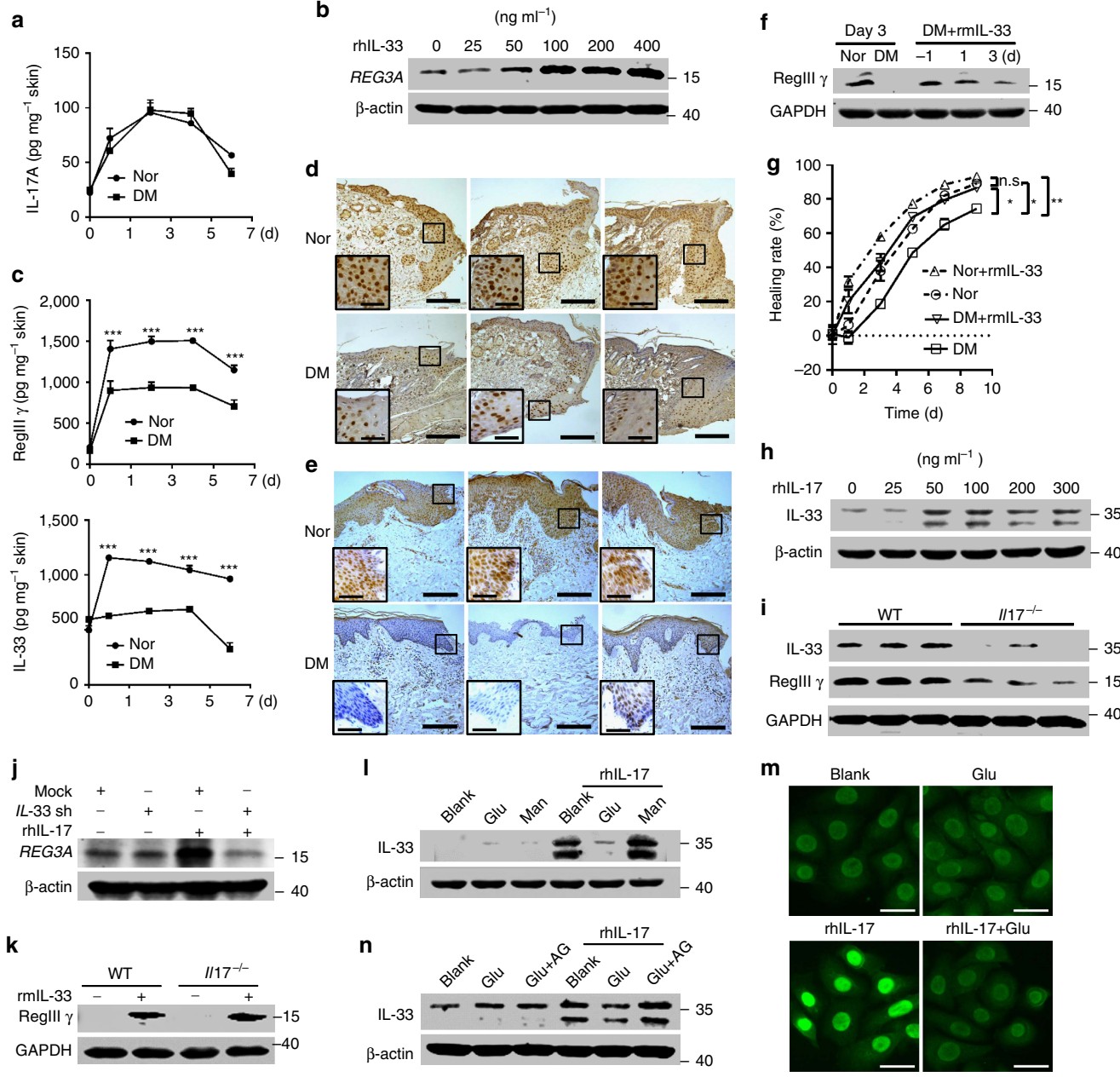

**Figure 2 | Hyperglycaemia inhibits IL-33 to decrease REG3A in diabetes.** (**a**) IL-17 production by ELISA in skin extracts taken from 2 mm skin surrounding the wound edges of normal and T1D mice at indicated times (n = 4). (**b**) Immunoblot of REG3A in NHEKs stimulated by different doses of rhIL-33 for 12 h. (**c**) RegIIIγ and IL-33 production by ELISA in skin extracts taken as in **a** (n = 4). (**d,e**) Immunohistochemical analyses of IL-33 in day-3 skin wounds of normal and T1D mice (**d**) or skin wounds of normal and diabetic patients taken as in Fig. 1b (**e**). Long scale bars represent 200 μm while short scale bars represent 50 μm. Black rectangles designate region of × 400 magnification shown in inset. (**f**) Immunoblot of RegIIIγ in the skin wounds of T1D mice treated with PBS or rmIL-33. The first two samples were day-3 skin wounds of normal and T1D mice. The other three samples were from the skin wounds of T1D mice treated with rmIL-33 a day before wounding (−1) or 1 or 3 days post wounding. (**g**) Wound healing of normal and T1D mice treated with (n = 7) or without rmIL-33 (n = 6). (**h**) IL-33 production in NHEKs induced by different doses of rhIL-17 for 24 h. (**i**) The production of IL-33 and RegIIIγ in day-3 skin wounds of wild-type (WT) and *Il17*[−/−] mice. (**j**) REG3A production in NHEKs induced by 200 ng ml[−1] rhIL-17 before and after IL-33 was silenced. (**k**) Immunoblot of RegIIIγ in skin wounds of WT and *Il17*[−/−] mice treated with PBS or 2 μg rmIL-33. (**l**) IL-33 production induced by 200 ng ml[−1] rhIL-17 in NHEKs exposed to 20 mM glucose or mannitol for 24 h. (**m**) Immunofluorescent staining of IL-33 in NHEKs treated as in **l**. Scale bars represent 25 μm. (**n**) IL-33 production in NHEKs treated with 200 ng ml[−1] rhIL-17 in the presence or absence of 20 mM glucose before and after AGE was inhibited by 2 mM aminoguanidine. The abbreviations used here are *IL-33* shRNA (*IL-33* sh), glucose (Glu), mannitol (Man), aminoguanidine (AG). *P < 0.05, **P < 0.01 and ***P < 0.001. P values were determined by two-way analysis of variance (ANOVA). Data are means ± s.e.m. and representative of two to three independent experiments.

Next we set out to determine what factor would inhibit IL-33 expression in diabetic skin wounds. Despite no reduction of IL-17 in the skin wounds of T1D mice, we found that recombinant human IL-17 (rhIL-17) induced IL-33 production in NHEKs (Fig. 2h, Supplementary Fig. 2g) and the deficiency of IL-17 decreased IL-33 production in the skin wounds (Fig. 2i, Supplementary Fig. 2h). Notably, IL-17 did induce REG3A/RegIIIγ production in keratinocytes (Fig. 2j, Supplementary Fig. 2i) and in skin wounds (Fig. 2i, Supplementary Fig. 2h). This induction was mediated by IL-33 as silencing IL-33 inhibited

rhIL-17-induced REG3A in NHEKs (Fig. 2j, Supplementary Fig. 2i) while the intradermal injection of rmIL-33 into the dorsal skin of $Il17^{-/-}$ mice before wounding restored RegIIIγ expression in the skin wounds (Fig. 2k, Supplementary Fig. 2j).

Because elevated glucose levels are characteristic of diabetes, and we also observed the higher glycated haemoglobin levels in T1D mice compared with the healthy controls (Supplementary Fig. 3a), we hypothesized that high glucose might participate in the regulation of IL-33 production induced by IL-17. Both neonatal and adult human epidermal keratinocytes were treated with mannitol (20 mM), normal (5.5 mM) or high (20 mM) glucose concentrations before stimulation with rhIL-17. High glucose inhibited the production of rhIL-17-induced IL-33 in both neonatal and adult human epidermal keratinocytes (Fig. 2l,m and Supplementary Fig. 3b,c), and this inhibition was dependent on glucose glycation as blocking the production of advanced glycation end products by aminoguanidine abrogated the inhibitory effect of high glucose on rhIL-17-induced IL-33 (Fig. 2n, Supplementary Fig. 3d). Consistent with the observation that IL-36γ was not decreased in the skin wounds of T1D mice, high glucose was unable to inhibit rhIL-17-induced IL-36γ in keratinocytes (Supplementary Fig. 3e). Collectively, these data demonstrate that hyperglycaemia inhibits IL-17 to induce IL-33 expression, thus leading to defective REG3A expression in the skin wounds of diabetic patients.

**Decreased REG3A exacerbates TLR3-induced inflammation.** Since REG3A inhibits wound-induced inflammation (Fig. 1f,g) and the expression of TLR3 was increased (Fig. 3a) to initiate inflammatory responses after skin injury[15], we next determined whether REG3A could regulate TLR3-induced inflammation in diabetic skin wounds. We first confirmed whether the expression of inflammatory cytokines in diabetic skin wounds would be dependent on the activation of TLR3. As expected, the significant reduction of TNF-α and IL-6 accompanied by improved wound healing was observed in the skin wounds of $Tlr3^{-/-}$ T1D mice compared with WT T1D mice (Fig. 3b,c). Consistent with the inhibitory effect of RegIIIγ on skin inflammation, the application of RegIIIγ also inhibited wound-induced TNF-α and IL-6 in WT T1D mice, but had no marked effect on the injury response in $Tlr3^{-/-}$ T1D mice (Fig. 3d). Moreover, TLR3 ligand, polyriboinosinic polyribocytidylic acid (poly(I:C)), induced both mRNA and protein of TNF-α and IL-6 in a time-dependent manner, and this increase was significantly inhibited by REG3A in NHEKs (Fig. 3e and Supplementary Fig. 4a,b). Although poly(I:C) also significantly induced interferon-β expression, REG3A did not inhibit this cytokine response (Supplementary Fig. 4c). Moreover, the inhibitory effect of REG3A on TLR3-induced pro-inflammatory cytokines was specific in keratinocytes as REG3A failed to inhibit poly(I:C)-induced TNF-α and IL-6 in macrophages (Supplementary Fig. 4d,e).

To determine the functional domain of REG3A involved in inhibiting TLR3-mediated pro-inflammatory cytokines, we generated several deletion mutants of REG3A (Supplementary Fig. 4f). The deletion of residues 40 to 175 of REG3A (N-REG3A), which encode CTLD domain, and the fragment containing residues 27 to 39 (REG3A-ΔCΔN), lost the inhibitory effect on poly(I:C)-induced TNF-α and IL-6 in NHEKs (Fig. 3f). Recombinant REG3A CTLD domain (C-REG3A) inhibited poly(I:C)-induced IL-6 and TNF-α as efficiently as full-length REG3A in both neonatal and adult human epidermal keratinocytes (Fig. 3f,g and Supplementary Fig. 4g). These data demonstrate that the CTLD domain is the functional domain of REG3A in the inhibition of TLR3-mediated inflammatory response.

**REG3A induces SHP-1 to inhibit TLR3 inflammation.** The finding that REG3A inhibits TLR3-mediated inflammation compelled us to further explore the mechanism by which REG3A limits TLR3 signalling. Multiple negative regulators including SHP-1, TRAF1, TNFAIP3 and SARM1 have been reported to limit TLR-induced inflammation[15,23]. Thus, we investigated whether these negative regulators were involved in TLR3-mediated signalling. Both REG3A and RegIIIγ markedly induced SHP-1 in a time-dependent manner in human keratinocytes or mouse skin (Fig. 4a–c, Supplementary Fig. 5a–c). In addition to full-length REG3A, the functional domain of REG3A (C-REG3A) induced SHP-1 production in dose- and time-dependent manners in NHEKs (Fig. 4d,e, Supplementary Fig. 5d,e). The expression of other negative regulatory genes such as SHP-2, TRAF1, TNFAIP3 and SARM were not induced (Supplementary Fig. 5f–i). As an inducer of REG3A, IL-33 also induced both mRNA and protein of SHP-1 and this induction was blocked after REG3A was silenced (Fig. 4f and Supplementary Fig. 5j–l). Moreover, the overexpression of SHP-1 significantly suppressed poly(I:C)-induced TNF-α and IL-6 (Fig. 4g), whereas silencing SHP-1 abrogated the inhibitory effect of REG3A on poly(I:C)-induced TNF-α and IL-6 in neonatal and adult human keratinocytes (Fig. 4h,i). Furthermore, the administration of SHP-1 inhibitor SSG into the mouse dorsal skin significantly increased the expression of TNF-α and IL-6 in the skin wounds of both WT normal and T1D mice (Fig. 4j). Thus, these results demonstrate that REG3A/RegIIIγ induces the negative regulator SHP-1 to control TLR3-mediated inflammation in skin wounds.

Next, we determined the signalling pathway involved in the induction of SHP-1 by REG3A/RegIIIγ. We first evaluated whether the activation of exostosin-like 3 (EXTL3), a receptor for REG3A (ref. 28), was required for REG3A to induce SHP-1. C-REG3A as well as full-length REG3A bound to EXTL3, and further mapping identified the region between residues 1 and 140, which encompasses the transmembrane region of EXTL3, as the C-REG3A binding site (Fig. 5a,b and Supplementary Fig. 6a). EXTL3 silencing significantly blocked the inhibitory effect of REG3A on poly(I:C)-induced TNF-α and IL-6, as well as inhibited REG3A-induced SHP-1 in NHEKs (Fig. 5c,d and Supplementary Fig. 6b,c). The inactivation of phosphatidylinositol 3 kinases (PI3K), AKT1/2 and signal transducer and activator of transcription 3(STAT3) but not other relevant signalling molecules blocked REG3A-induced SHP-1 production (Fig. 5e, Supplementary Fig. 6d). To confirm PI3K is the downstream signal molecule for EXTL3, we checked whether REG3A activated PI3K to induce AKT phosphorylation after EXTL3 was silenced. REG3A increased AKT phosphorylation and this increase was inhibited after EXTL3 was silenced (Fig. 5f, Supplementary Fig. 6e). Moreover, AKT1/2 inhibitor dampened REG3A-induced STAT3 phosphorylation (Fig. 5g, Supplementary Fig. 6f) while STAT3 inhibitor did not prevent AKT phosphorylation induced by REG3A (Fig. 5h, Supplementary Fig. 6g). These results suggest that REG3A activates the EXTL3–PI3K–AKT–STAT3 signalling pathway to induce SHP-1 expression in keratinocytes.

**REG3A inhibits TLR3-activated JNK2.** To gain a better understanding of the role of SHP-1 in TLR3 signalling, we next dissected TLR3-activated downstream signalling. The treatment of NHEKs with multiple inhibitors in the presence or absence of poly(I:C) demonstrated that c-Jun N-terminal kinase (JNK) inhibitor, but not other inhibitors, significantly inhibited the mRNA expression of poly(I:C)-induced both TNF-α and IL-6 (Supplementary Fig. 7a). Consistent with decreased mRNA, JNK inhibitor significantly inhibited the protein production of poly(I:C)-induced TNF-α and IL-6 in both neonatal and adult

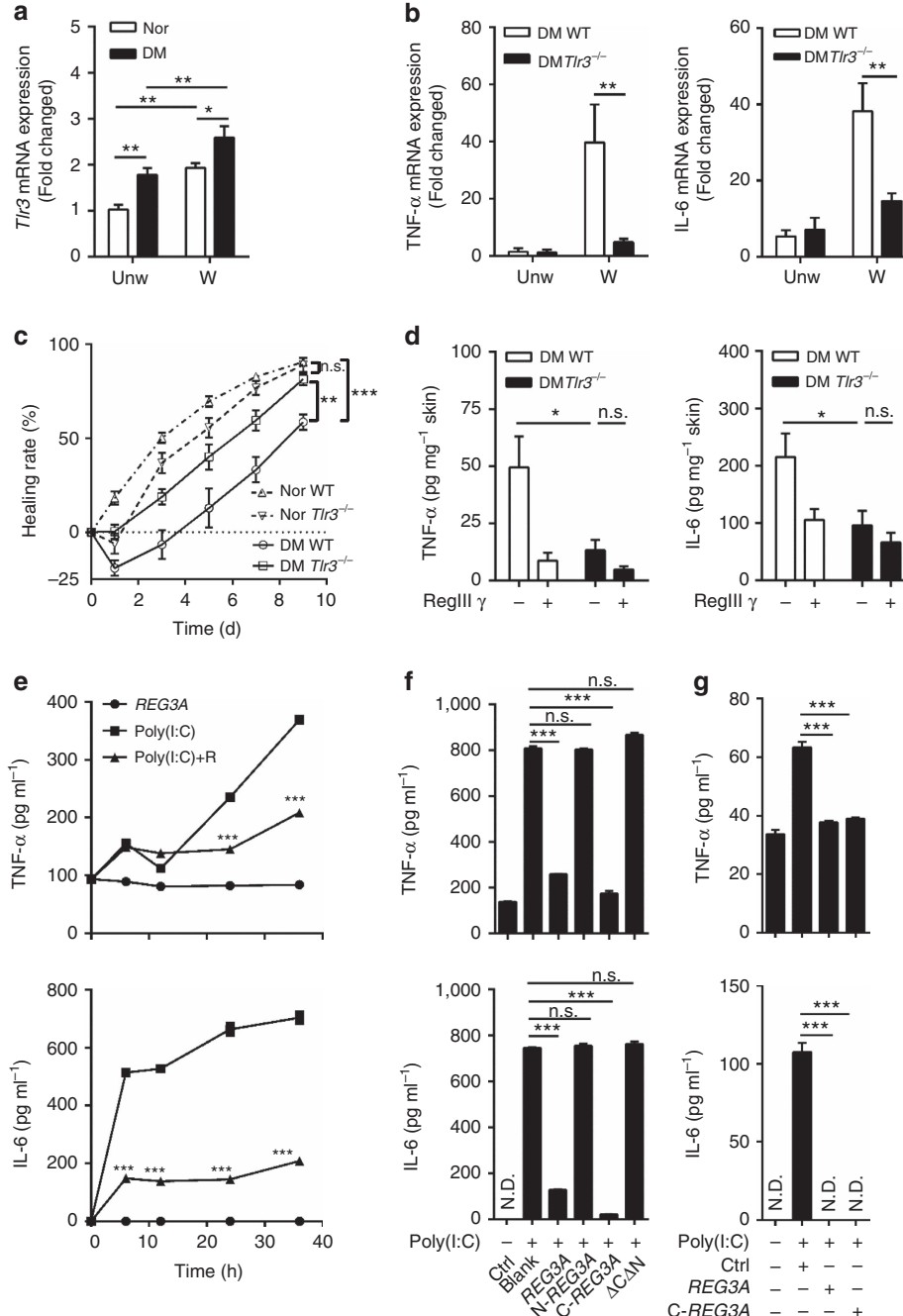

**Figure 3 | REG3A inhibits TLR3-induced TNF-α and IL-6. (a)** Quantification of *Tlr3* mRNA expression in day-3 skin wounds of normal and T1D mice ($n = 6$). **(b)** Quantification of TNF-α and IL-6 mRNA expression in the skin extracts taken from 2 mm surrounding the wound edges of WT and *Tlr3*$^{-/-}$ T1D mice at day 3 ($n = 6$). **(c)** Wound healing of normal WT ($n = 12$), normal *Tlr3*$^{-/-}$ ($n = 8$), diabetic WT ($n = 12$) and diabetic *Tlr3*$^{-/-}$ ($n = 10$) mice. **(d)** Quantification of TNF-α ($n = 5$) and IL-6 ($n = 8$) protein by ELISA in the skin extracts taken from 2 mm surrounding the wound edges of WT and *Tlr3*$^{-/-}$ T1D mice treated with or without RegIIIγ. **(e)** Quantification of TNF-α and IL-6 protein by ELISA in NHEKs treated with 5 μg ml$^{-1}$ poly(I:C) in the presence or absence of 30 nM REG3A ($n = 3$). **(f)** Quantification of TNF-α and IL-6 protein by ELISA in NHEKs treated with 5 μg ml$^{-1}$ poly(I:C) in the presence or absence of different mutants of REG3A ($n = 3$). **(g)** Quantification of TNF-α and IL-6 protein by ELISA in adult human epidermal keratinocytes (AHEKs) treated with 10 μg ml$^{-1}$ poly(I:C) in the presence or absence of 30 nM REG3A or C-REG3A ($n = 3$). $^*P < 0.05$, $^{**}P < 0.01$ and $^{***}P < 0.001$. NS, no significance; ND, not detected. $P$ values were analysed by one-way analysis of variance (ANOVA; **f,g**) or two-way ANOVA (**a–e**). Data are means ± s.e.m. and representative of two to three independent experiments.

human keratinocytes (Fig. 6a,b). To confirm that JNK is the key downstream signal molecule for TLR3, we then check whether the activation of TLR3 could induce JNK phosphorylation. Poly(I:C) markedly increased the phosphorylation of JNK, and this increase was eliminated by JNK inhibitor in NHEKs (Fig. 6c, Supplementary Fig. 7b). In addition to JNK, the phosphorylation

of c-Jun, one key downstream molecule of JNKs, was markedly inhibited by JNK inhibitor in NHEKs (Fig. 6c, Supplementary Fig. 7b). Moreover, the overexpression of c-Jun significantly increased the luciferase activities of TNF-α and IL-6 promoters, whereas this induction was attenuated when c-Jun binding sites were mutated in TNF-α and IL-6 promoters (Fig. 6d,e).

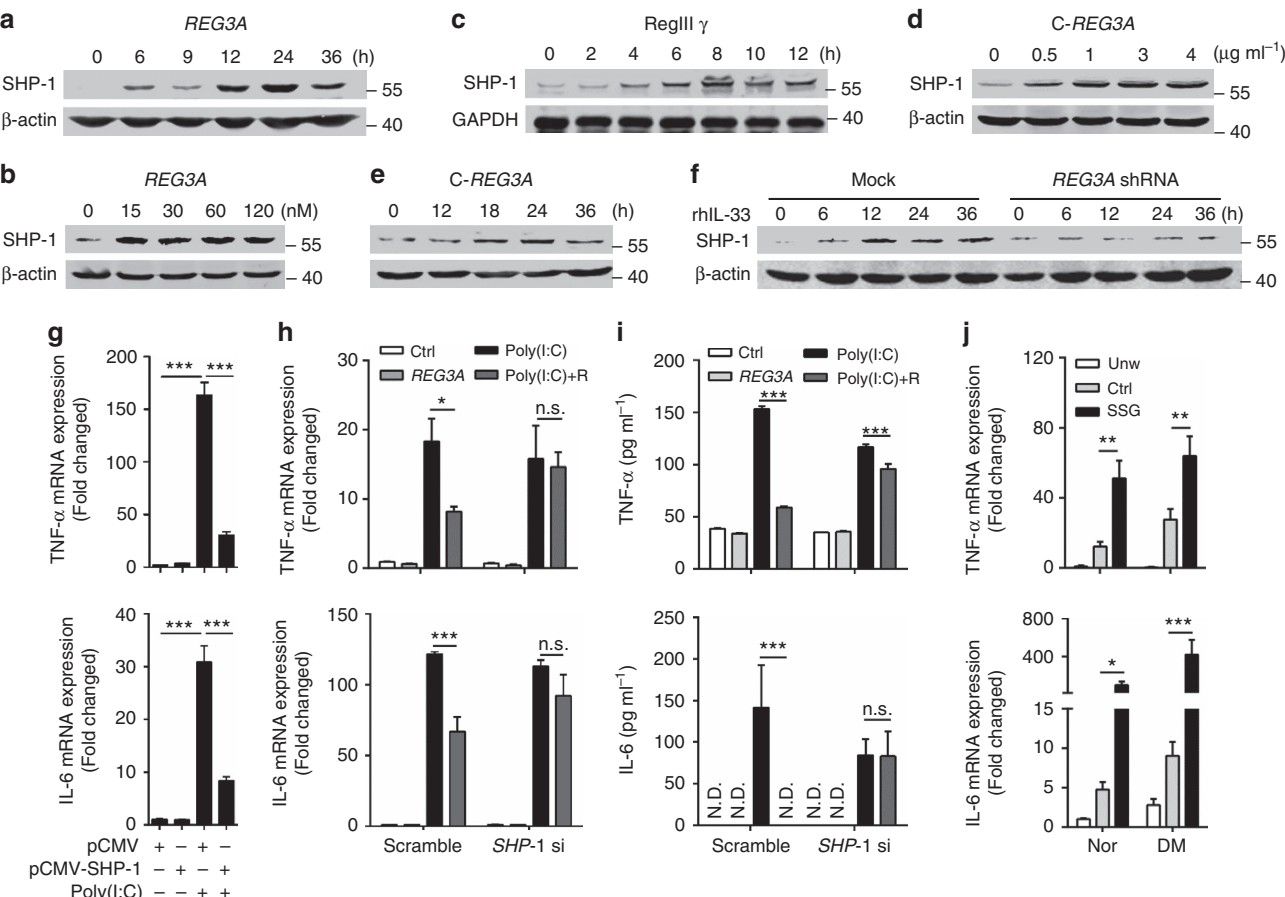

**Figure 4 | REG3A induces SHP-1 to inhibit TLR3-induced TNF-α and IL-6.** (**a**) Immunoblot of SHP-1 in NHEKs stimulated with 30 nM REG3A for indicated times. (**b**) Immunoblot of SHP-1 in AHEKs stimulated with different doses of REG3A for 24 h. (**c**) Immunoblot of SHP-1 in mouse skin intradermally injected with 100 µg RegIIIγ for indicated times. (**d**,**e**) Immunoblot of SHP-1 in NHEKs transfected with different doses of pSG5-C-REG3A (**d**) for indicated times (**e**). (**f**) Immunoblot of SHP-1 in NHEKs stimulated with 100 ng ml$^{-1}$ rhIL-33 before and after REG3A silencing. (**g**) TNF-α and IL-6 mRNA expression in NHEKs transfected with 1 µg pCMV vector or pCMV-SHP-1 in the presence or absence of 5 µg ml$^{-1}$ poly(I:C) for 24 h (n = 3). (**h**) TNF-α and IL-6 mRNA expression in NHEKs treated with 5 µg ml$^{-1}$ poly(I:C) and/or 30 nM REG3A before or after SHP-1 silencing (n = 3). (**i**) TNF-α and IL-6 production in AHEKs treated as in **h** (n = 3). (**j**) TNF-α and IL-6 mRNA expression in day-3 skin wounds of WT normal or T1D mice (n = 5). The mice were intradermally injected with $H_2O$ or 4 mg of SHP-1 inhibitor SSG before wounding. Three days later, 2 mm skin surrounding the wounds was taken for RNA isolation. The abbreviations used here are SHP-1 shRNA (SHP-1 sh) and REG3A (R). *P < 0.05, **P < 0.01 and ***P < 0.001. NS, no significance; ND, not detected. P values were analysed by one-way analysis of variance (ANOVA; **g**) or two-way ANOVA (**h**–**j**). Data are means ± s.e.m. and representative of two to three independent experiments.

Because JNK1 and JNK2 are two ubiquitously expressed isoforms in many cell types[29], we then isolated primary murine keratinocytes from WT, Jnk1$^{-/-}$ and Jnk2$^{-/-}$ newborn mice to further determine which JNK was the downstream molecule involved in TLR3 signalling in keratinocytes. Poly(I:C) induced the phosphorylation of JNK2 as well as c-Jun in WT and Jnk1$^{-/-}$ murine keratinocytes (Fig. 6f,g and Supplementary Fig. 7c,d), but failed to induce the phosphorylation of JNK1 or c-Jun in Jnk2$^{-/-}$ murine keratinocytes (Fig. 6h, Supplementary Fig. 7e), suggesting that JNK2 is a key downstream molecule of TLR3. Consistent with the in vitro observation, the phosphorylation of JNK2 was increased in skin wounds of WT normal and T1D mice, but decreased in the skin wounds of Tlr3$^{-/-}$ normal and T1D mice (Fig. 6i,j and Supplementary Fig. 7f). Moreover, the deficiency of JNK2 significantly decreased the production of TNF-α and IL-6 in the skin wounds of normal and T1D mice (Fig. 6k). Notably, wound closure in Jnk2$^{-/-}$ but not Jnk1$^{-/-}$ normal and T1D mice was significantly accelerated (Fig. 6l and Supplementary Fig. 7g). In line with the inhibitory effect of REG3A on TLR3-mediated inflammatory responses, REG3A and its

functional domain C-REG3A inhibited poly(I:C)-activated JNK2 phosphorylation in neonatal and adult human keratinocytes (Fig. 6m–o, Supplementary Fig. 7h–j). Collectively, these data show that REG3A acts on TLR3–JNK2 pathway to control the magnitude of inflammation in skin wounds.

**SHP-1 mediates REG3A inhibition of TLR3-activated JNK2.** Having established the involvement of SHP-1 in the suppression of keratinocyte cytokine release, and the role of REG3A/RegIIIγ in the inhibition of TLR3-activated JNK2 phosphorylation, we then determined whether REG3A inhibited JNK2 phosphorylation via SHP-1. Poly(I:C) markedly increased JNK2 phosphorylation, while this increase was inhibited by the overexpression of SHP-1 but not SHP-1 with a mutation at cysteine 453 in keratinocytes (Fig. 7a, Supplementary Fig. 8a). SHP-1 silencing also blocked the inhibitory effect of REG3A on poly(I:C)-induced JNK2 phosphorylation (Fig. 7b, Supplementary Fig. 8b). Furthermore, SHP-1 exerted its phosphatase activity via the interaction with JNK2 as demonstrated by

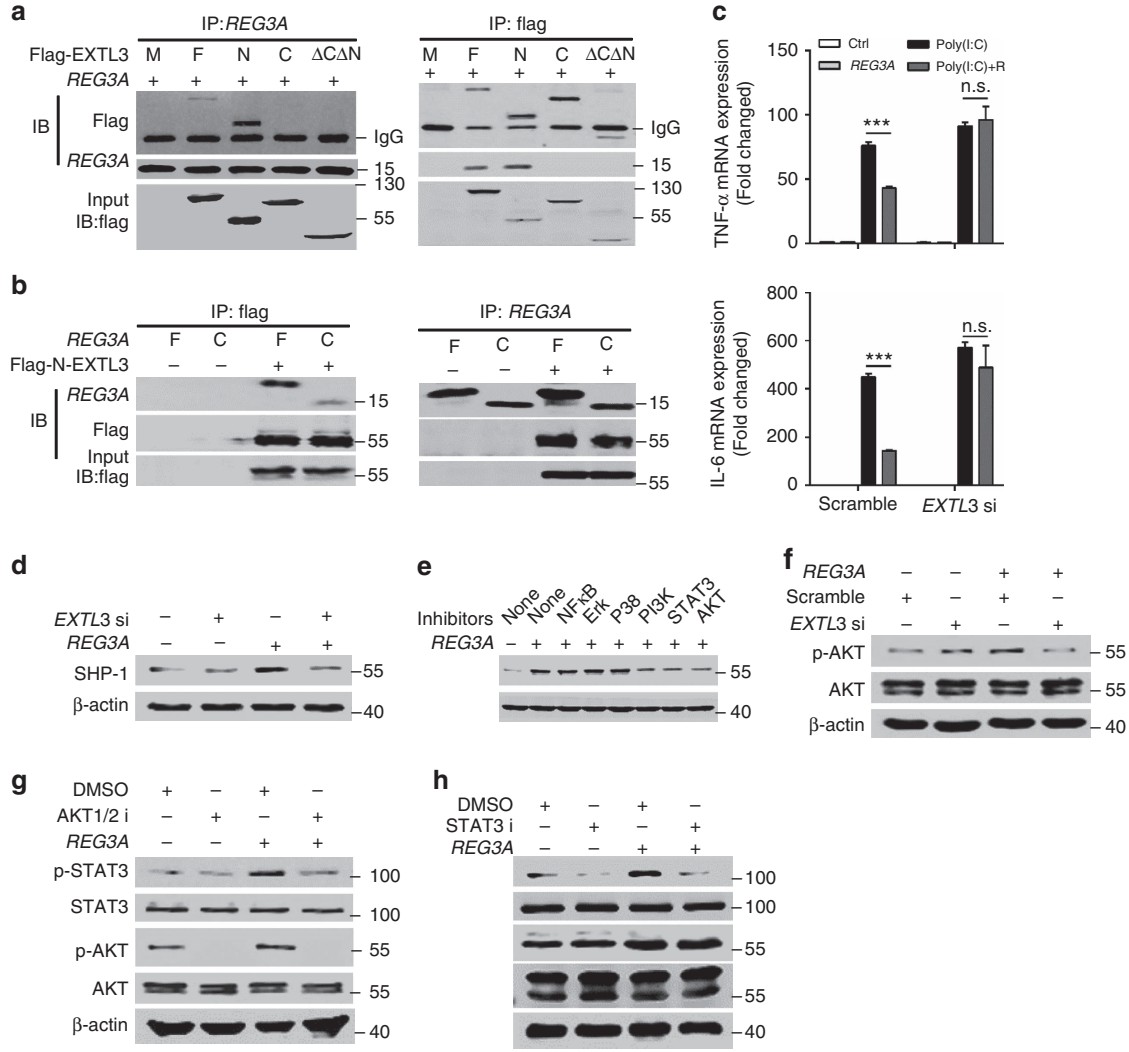

**Figure 5 | REG3A activates EXTL3-AKT-STAT3 to induce SHP-1.** (**a**) Interaction between Flag-tagged different domains of EXTL3 and REG3A assessed by immunoblot analysis after immunoprecipitation with anti-Flag or anti-REG3A. M: mock; F: full-length EXTL3; N: N-EXTL3 (1–548); C: C-EXTL3 (141–919); ΔNΔC: EXTL3 ΔNΔC (141–548). (**b**) Interaction between REG3A or C-REG3A and Flag-tagged N-EXTL3 assessed by immunoblot analysis after immunoprecipitation with anti-REG3A or anti-Flag. F: full-length REG3A; C: CTLD domain of REG3A. (**c**) Quantification of TNF-α and IL-6 mRNA expression in NHEKs stimulated with 5 μg ml$^{-1}$ poly(I:C) and/or 30 nM REG3A before or after EXTL3 silencing (n = 3). (**d**) Immunoblot of SHP-1 in NHEKs treated with 30 nM REG3A before or after EXTL3 silencing. EXTK3 si: EXTL3 siRNA. (**e**) SHP-1 production in NHEKs treated with 30 nM REG3A in the presence or absence of NF-κB inhibitor (Bay11, 10 μM), Erk inhibitor (PD98059, 20 μM), p38 MAPK inhibitor (SB202190, 5 μM), PI3K inhibitor (LY294002, 50 μM), STAT3 inhibitor (S3I-201,50 μM) and AKT inhibitor (AKT1/2 inhibitor, 8 μM). (**f**) Immunoblot of phosphorylated AKT in NHEKs treated with 30 nM REG3A for 1h before or after EXTL3 was silenced. (**g**) Immunoblot of p-STAT3 and p-AKT in NHEKs treated with 30 nM REG3A and/or AKT1/2 inhibitor for 1h. AKT1/2 i: AKT1/2 inhibitor. (**h**) Immunoblot of p-STAT3 and p-AKT in NHEKs treated with 30 nM REG3A and/or STAT3 inhibitor (S3I-201) for 1h. STAT3 i: STAT3 inhibitor. ***P < 0.001. NS, no significance. P values were analysed by two-way analysis of variance (ANOVA). Data are means ± s.e.m. and representative of two to three independent experiments.

immunoprecipitation of REG3A- and poly(I:C)- treated keratinocytes with SHP-1 antibody and detection with antibody to JNK2, or when JNK2 was precipitated and SHP-1 antibody was used for detection (Fig. 7c).

Next, we confirmed the physiological relevance of SHP-1 in inhibiting JNK2 activation to control inflammatory responses necessary for optimal wound healing. In parallel with increased RegIIIγ, SHP-1 production was markedly elevated, but JNK2 phosphorylation was decreased in the skin wounds of normal mice (Fig. 7d, Supplementary Fig. 8c). In contrast to normal mice, the production of RegIIIγ and SHP-1 was decreased while the phosphorylation of JNK2 was increased in the skin wounds of T1D mice (Fig. 7d, Supplementary Fig. 8c), which was

accompanied by increased production of TNF-α and IL-6 (Supplementary Fig. 8d). Furthermore, the blockade of RegIIIγ by RegIIIγ-neutralizing antibody decreased SHP-1 production but increased JNK2 phosphorylation in normal mice (Fig. 7e, Supplementary Fig. 8e) while the administration of RegIIIγ before wounding induced SHP-1 production but inhibited JNK2 phosphorylation in skin wounds of T1D mice (Fig. 7f, Supplementary Fig. 8f). To further confirm that REG3A induces SHP-1 to promote wound healing, we evaluated the healing process after SHP-1 was inhibited. REG3A restored the healing process of T1D mice back to that of normal mice, but the administration of SHP-1 inhibitor significantly inhibited this process (Fig. 7g). All these data confirm that defective expression

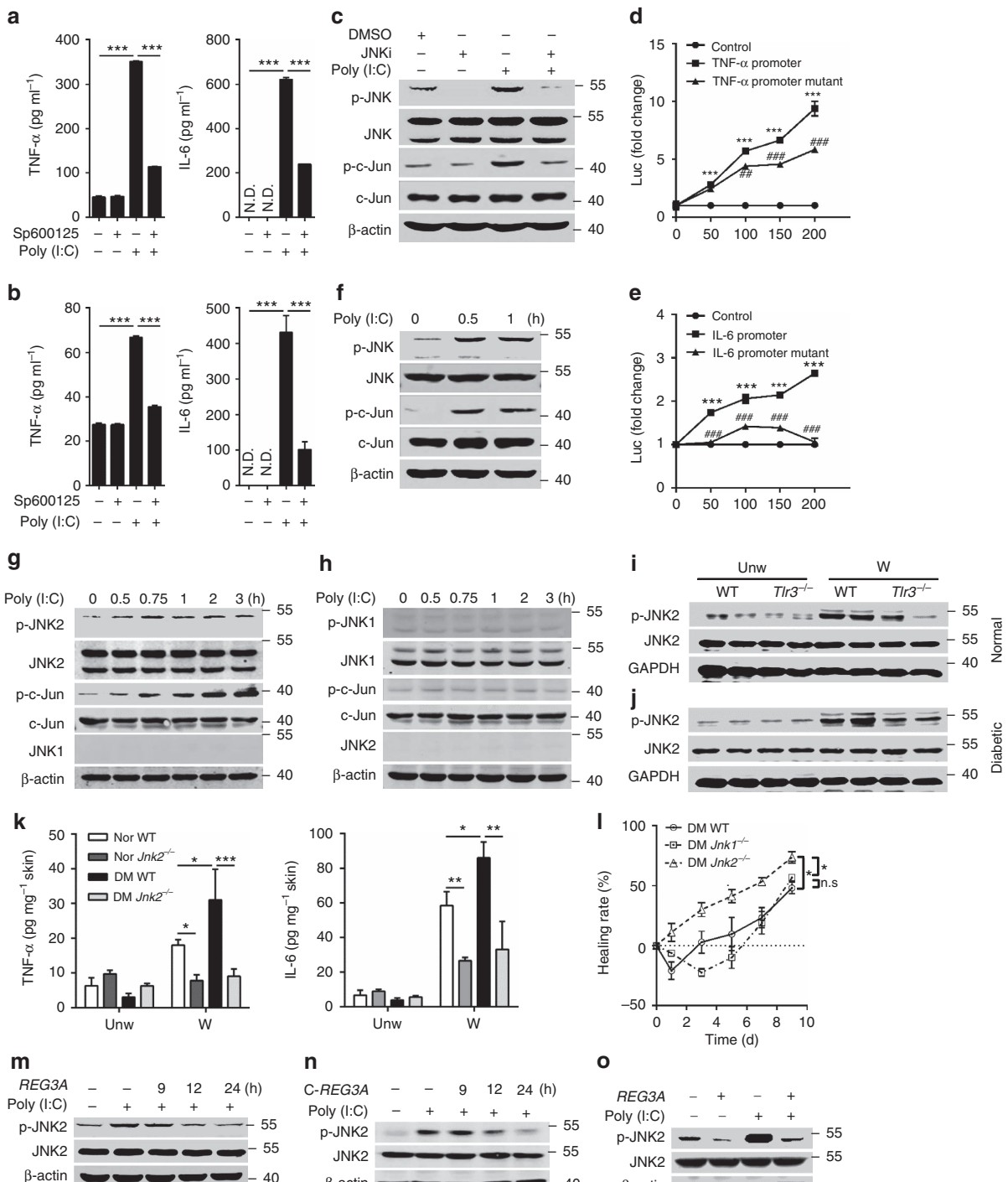

**Figure 6 | TLR3 specifically activates JNK2 in response to injury.** (**a,b**) Quantification of TNF-α and IL-6 production by ELISA in NHEKs treated with 5 μg ml⁻¹ poly(I:C) (**a**) or in AHEKs treated with 10 μg ml⁻¹ poly(I:C) (**b**) in the presence or absence of JNK inhibitor (SP600125, 15 μM) for 24 h (n = 3). (**c**) The phosphorylation of JNK and c-Jun in NHEKs stimulated by 5 μg ml⁻¹ poly(I:C) in the presence or absence of JNK inhibitor for 1 h. JNK i: JNK inhibitor. (**d,e**) Normalized luciferase activity in 293T cells co-transfected with the plasmid expressing c-Jun (0, 50, 100, 150, 200 ng) and plasmids containing TNF-α promoter (50 ng; **d**), or TNF-α promoter with mutation in c-Jun binding site (50 ng; **d**), or IL-6 promoter (100 ng; **e**), or IL-6 promoter with the mutation in c-Jun binding site (100 ng; **e**) for 24 h. (**f**) The phosphorylation of JNK and c-Jun in WT primary murine keratinocytes stimulated by 10 μg ml⁻¹ poly(I:C) for indicated times. (**g**) The phosphorylation of JNK2 and c-Jun in $Jnk1^{-/-}$ primary murine keratinocytes stimulated by 10 μg ml⁻¹ poly(I:C) for indicated times. (**h**) The phosphorylation of JNK1 and c-Jun in $Jnk2^{-/-}$ primary murine keratinocytes stimulated by 10 μg ml⁻¹ poly(I:C) for indicated times. (**i,j**) JNK2 phosphorylation in the skin wounds of WT and $Tlr3^{-/-}$ normal mice (**i**) or T1D mice (**j**). (**k**) Quantification of TNF-α and IL-6 production by ELISA in day-3 skin wounds of WT and $Jnk2^{-/-}$ normal and T1D mice (n = 6). (**l**) Wound healing of WT (n = 6), $Jnk1^{-/-}$ (n = 4) and $Jnk2^{-/-}$ T1D mice (n = 6). (**m,n**) JNK2 phosphorylation in NHEKs stimulated by 5 μg ml⁻¹ poly(I:C) in the presence or absence of 30 nM REG3A (**m**) or C-REG3A (**n**). (**o**) JNK2 phosphorylation in AHEKs stimulated by 10 μg ml⁻¹ poly(I:C) in the presence or absence of 30 nM REG3A. *$P < 0.05$, ** or ##$P < 0.01$ and ***or ###$P < 0.001$. P value was analysed by one-way analysis of variance (ANOVA; **a,b**) or two-way ANOVA (**d,e,k,l**). Data are the means ± s.e.m. and representative of two to three independent experiments.

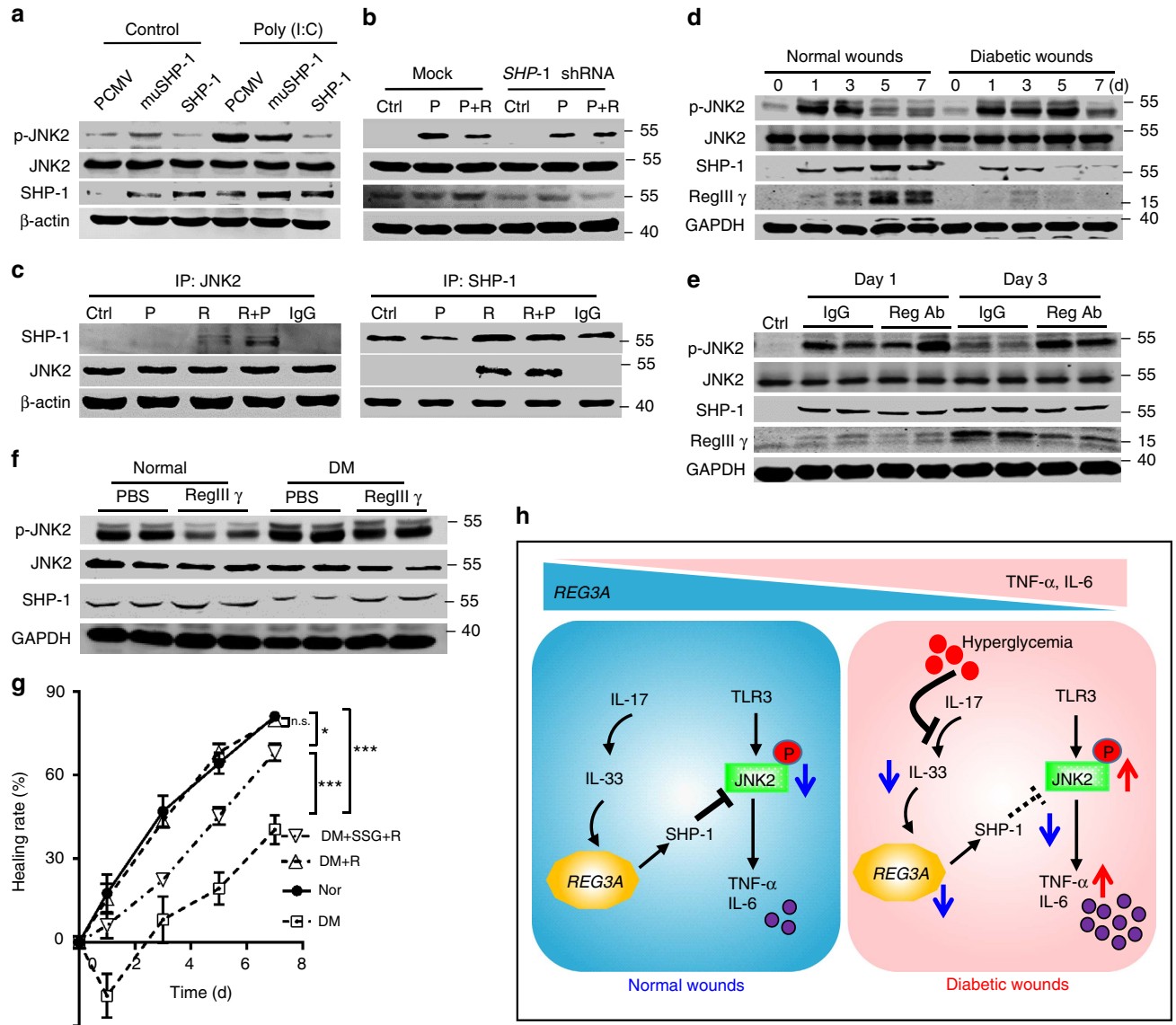

**Figure 7 | SHP-1 dephosphorylates phospho-JNK2 to regulate wound healing.** (**a**) Phosphorylated JNK2 in NHEKs transfected with pCMV vector, pCMV-muSHP-1(C453S) and pCMV-SHP-1 in the presence or absence of $5\,\mu g\,ml^{-1}$ poly(I:C). (**b**) Phosphorylated JNK2 in NHEKs stimulated with $5\,\mu g\,ml^{-1}$ poly(I:C) and/or 30 nM REG3A before or after SHP-1 silencing. (**c**) Interaction between SHP-1 and JNK2 in NHEKs stimulated with $5\,\mu g\,ml^{-1}$ poly(I:C) and/or 30 nM REG3A assessed by immunoblot analysis after immunoprecipitation with anti-SHP-1 or anti-JNK2. (**d**) Immunoblot of p-JNK2, SHP-1 and RegIIIγ in the skin extracts taken from 2 mm surrounding the wound edges of WT normal and T1D mice at indicated times. (**e**) Immunoblot of p-JNK2, SHP-1 and RegIIIγ in the skin wounds of WT normal mice injected with IgG or RegIIIγ-neutralizing antibody. (**f**) Immunoblot of p-JNK2 and SHP-1 in the skin wounds of WT normal and T1D mice injected with 100 μg RegIIIγ. (**g**) Wound healing in T1D mice treated with or without 100 μg RegIIIγ before or after SHP-1 was inhibited by its inhibitor SSG ($n=9$). (**h**) The schematic graph reflects the interaction between REG3A and TLR3 signalling. After skin injury, IL-33 induces REG3A expression in epidermal keratinocytes. REG3A, in turn, acts on keratinocytes to induce the negative regulator SHP-1 to selectively inhibit TLR3-activated JNK2, thus controlling TLR3-induced inflammation in skin wounds. However, in diabetes, hyperglycaemia inhibits IL-33 expression induced by IL-17. The reduction of IL-33 leads to the decrease in REG3A and SHP-1 but increased TLR3-activated JNK2 phosphorylation, thus exacerbating inflammation in diabetic skin wounds. The abbreviations used here are control (Ctrl), poly(I:C) (P), REG3A (R) and Reg Ab (RegIIIγ antibody). $*P<0.05$, and $***P<0.001$. NS, no significance. $P$ value was analysed by two-way analysis of variance (ANOVA). Data are the means ± s.e.m. and representative of two independent experiments.

of RegIIIγ dysregulates TLR3-mediated inflammatory response via decreasing SHP-1 to increase JNK2 phosphorylation in diabetic skin wounds.

## Discussion
In the absence of infection, excessive production of inflammatory cytokines during wound repair is undesirable and may impair wound healing responses, especially in the context of metabolic disorders, such as diabetes. REG3A has been proposed to regulate uncontrolled inflammatory responses in ulcerative colitis[27], but was not known to be involved in the regulation of inflammatory responses after skin injury. Here we observed that defective REG3A expression may impair wound healing in diabetes through the exacerbation of TLR3-mediated inflammation. Our results reveal that REG3A, induced by IL-33, inhibits TLR3-mediated inflammation in skin wounds. The mechanism for REG3A-inhibited inflammatory response involves

induction of the negative regulatory factor SHP-1, which exhibits a major role in delayed wound healing by inhibiting TLR3-activated JNK2 phosphorylation. The reduction of cutaneous REG3A, possibly as a consequence of decreased IL-17-induced IL-33 caused by hyperglycaemia, will therefore lead to excessive production of TLR3-mediated pro-inflammatory cytokines observed in diabetes after skin trauma (Fig. 7h). Thus, the identification of REG3A as a mediator for inflammatory responses in skin wounds, and the elucidation of its mechanism of action, provides crucial information for understanding the process of wound healing in diabetes. These findings also offer potential drug or targets for therapy.

An appropriate inflammatory response is fundamentally critical for normal wound repair responses. Without this appropriate inflammatory response, wound healing is delayed and the host is more susceptible to microbial invasion[4,30]. Therefore, normal immune defense requires that a sensitive balance is maintained to minimize unnecessary inflammation, yet rapidly respond to infection and injury[15]. Here we show that this balance is impaired at the epithelial surface in the context of diabetes, and that innate immune system uses REG3A to enable the control of this balance during wound healing. Under normal conditions, the activation of TLR3–JNK2 signalling leads to local release of inflammatory cytokines, and REG3A induces SHP-1 to specifically modulate the expression of TLR3-induced pro-inflammatory cytokines, such as TNF-α and IL-6, but not TLR3-induced IFNβ. Thus, this prevents excessive or unwanted inflammation, while maintaining the important role in clearing pathogens during wound repair. However, in the context of diabetes, the expression of REG3A was impaired in skin wounds and SHP-1 was observed to be diminished in parallel, thus leading to excessive production of TNF-α and IL-6 and delayed wound healing. Therefore, the appropriate REG3A expression is essential for tissue homeostasis after skin injury. Moreover, our previous observation has demonstrated that REG3A can regulate keratinocyte proliferation and differentiation to promote wound re-epithelialization[28]. The capacity of REG3A to control TLR3-induced inflammation and its effect on keratinocyte proliferation are critical, but irreplaceable, for normal wound healing responses in diabetes, as SHP-1silencing in vitro did not inhibit keratinocyte proliferation induced by REG3A itself (Supplementary Fig. 9), and the inactivation of SHP-1 by SSG in vivo only partially inhibited RegIIIγ-promoted wound healing in diabetes. Both of these functions of REG3A link this system to essential processes involved in wound healing and suggest that REG3A could be a potential drug for the treatment with delayed or unhealed skin wounds of patients with diabetes.

Multiple factors including cytokines, transcription factors and bacteria are involved in the induction of REG3A (refs 31–33). In our previous studies, we have screened multiple cytokines including TNF-α, IL-6, IFN-γ, IL-13, IL-12, IL-17 and IL-22 that are elevated in skin wounds, and showed that IL-17 was an important molecule to induce REG3A expression in keratinocyte[28]. Here we have also confirmed that IL-17 did induce REG3A expression in keratinocytes. However, in skin wounds of diabetic mice, IL-17 was not decreased in a similar fashion as RegIIIγ, suggesting that IL-17 was not a cytokine that led to decreased REG3A in diabetic skin wounds. Although we further screened multiple cytokines and found that both IL-33 and IL-36γ induced REG3A expression in keratinocytes, only IL-33 was decreased as similar as RegIIIγ in diabetic skin wounds, which was consistent with the previous report that IL-33 was decreased in diabetic skin wounds[34]. We then confirmed that decreased REG3A expression was possibly due to decreased IL-33 in diabetic skin wounds, as the administration of IL-33 into the dorsal skin of diabetes restored RegIIIγ expression. To our surprise, the further study showed that the production of IL-33 was induced by IL-17 in keratinocytes, even though IL-17 production was not decreased in the skin wounds of diabetic mice. Therefore, a question arose as to what inhibited IL-17 to induce IL-33 in keratinocytes. It is well known that intracellular glucose levels and glucose glycation are frequently high in diabetics[35,36]. We thereby assumed that high glucose might influence IL-33 in response to IL-17. Our results clarify the role of high glucose in the inhibition of IL-17-induced IL-33 expression via glucose glycation in keratinocytes and reveal that hyperglycaemia is a possible causative mechanism of decreased REG3A by IL-33 in diabetic skin wounds.

Although it has been implicated that REG3A can regulate inflammation in gut, the underlying mechanism by which REG3A regulates inflammatory responses in skin wounds was poorly explored. Here we show that REG3A induces the negative regulator SHP-1 to inhibit TLR3-activated JNK2 phosphorylation, thus limiting TLR3 signalling. Previous reports have demonstrated that SHP-1 increases TLR- and RIG-I- activated type I interferon by directly binding to the kinase tyrosyl-based inhibitory motif (KTIM) of interleukin-1 receptor-associated kinase 1 (IRAK1) for inactivating IRAK1 activity[37,38], whereas suppresses TLR-activated pro-inflammatory cytokine production by inhibiting TRAF6 ubiquitination via translocated intimin receptor (Tir)[39]. Besides SHP-1, SHP-2 inhibits TLR3-activated inflammatory response by directly binding to TANK binding kinase (TBK1)[40]. Despite the importance of SHP-1 and SHP-2 in TLR signalling, how SHP-1 and SHP-2 are regulated is less clear. In our system, we found that REG3A induces SHP-1 but not SHP-2 in keratinocytes. More importantly, we found that SHP-1induced by REG3A/RegIIIγ bound to JNK2, but not previously reported IRAK1 (ref. 38) or TRAF6 (ref. 39), and then inactivated JNK2 phosphorylation in keratinocytes. One explanation will be that RER3A/RegIIIγ specifically inhibited TLR3-activated pro-inflammatory cytokines such as TNF-α and IL-6, but not IFNβ, as IRAK1 is involved in the regulation of type I interferon production downstream of TLR3 (ref. 38). Alternatively, JNK2 has its own KTIM for SHP-1 recognition, but TRAF6 lacks this motif[37]. Therefore, SHP-1 binds to JNK2, a key downstream signal molecule of TLR3 in the induction of pro-inflammatory cytokines, leading to a decreased pro-inflammatory response. This observation reveals that the regulation of JNK2 activity would be an alternative strategy for control of excessive inflammation in diabetic skin wounds.

Taken together, these findings support our discovery that REG3A/ RegIIIγ is important for the control of inflammation after skin injury. The aberrant expression of REG3A amplifies inflammation in diabetic skin wounds and may be a previously unknown key element in the pathogenesis of delayed or unhealed wounds of diabetic individuals. The present finding, together with our previous observation that REG3A regulates keratinocyte proliferation and differentiation to promote wound healing, implicates the potential of REG3A as a therapeutic target in wound healing. The identification of REG3A functions in skin provides novel insights into pathways contributing to wound repair in diabetes and may ultimately lead to the development of immunomodulatory strategies for the treatment with diabetic skin wounds.

## Methods
**Human samples.** All human skin samples were obtained from five normal patients and five diabetic patients with acute injury in Changhai Hospital, Second Military Medical University, China. Patient information is listed in Supplementary Table 1. Five-millimetre skin surrounding wound edges of normal and diabetic patients was taken for immunohistochemical analysis of REG3A or IL-33, while 2 mm skin

surrounding wound edges was taken for RNA isolation. All human sample acquisitions were approved by the committee on ethics of biomedicine, Second Military Medical University, and performed in accordance with the Declaration of Helsinki Principles. All the participants provided written informed consent, which was obtained before enrolment in the study.

**Animals.** $Tlr3^{-/-}$ C57BL/6 breeding pairs (Stock No: 009675) were purchased from Jackson Laboratory. $Jnk1^{-/-}$, $Jnk2^{-/-}$ and $Il17^{-/-}$ C57BL/6 breeding pairs were provided by Drs Xin Lin and Chen Dong from Tsinghua University, respectively. All the mice were housed and bred in specific pathogen-free conditions in the animal facilities in East China Normal University. All mouse experiments were approved by East China Normal University Animal Care and Use Committee. All the surgeries were performed under anaesthesia and all the efforts were made to minimize suffering. For all the mouse studies, we performed preliminary experiments to determine requirements for sample size. The mice were assigned randomly to experimental groups but not performed in a blinded manner. All the mice used in this study were 7-week to10-week old. Both male and female mice were used for the experiments. The male and female mice were group-housed with littermates.

**STZ-induced diabetic (T1D) mice.** Age-matched WT C57BL/6 mice, $Jnk1^{-/-}$, $Jnk2^{-/-}$, $Tlr3^{-/-}$ C57BL/6 mice (7–8 weeks) were used to induce diabetes. Fifty milligrams per kg body weight of STZ was injected intraperitoneally into the mice for 5 days. After 10 days, the blood glucose levels were measured by using glucose meters (Jinque, China). The mice were considered diabetic if the non-fasted gly-caemia was higher than 20 mM.

**Excisional wound model.** Dorsal mouse fur from age-matched control or $Tlr3^{-/-}$, $Jnk1^{-/-}$, $Jnk2^{-/-}$ and $Il17^{-/-}$ C57BL/6 mice (8–10 weeks) was removed by using chemical depilation (Veet) under isoflurane anaesthesia. Excisional dorsal skin wounds were made with an 8 mm sterile biopsy punch. The mice were killed on day 1, 3, 5, 7 post wounding, and 5 mm skin surrounding wound edges was used for immunohistochemical analysis, 2 mm skin surrounding wound edges was collected for quantitative PCR, western blot or ELISA (enzyme-linked immunosorbent assay). For wound healing assay, the wound areas were photographed every other day and calculated by Image J.

**Cutaneous administration of recombinant proteins or the inhibitor.** After mouse fur was removed, 100 μg of RegIIIγ or PB control (100 μl) was intradermally injected into dorsal skin of age-matched control or $Tlr3^{-/-}$ C57BL/6 normal and T1D mice, or 2 μg of rmIL-33 was intradermally injected into the dorsal skin of age-matched control and $Il17^{-/-}$ C57BL/6 normal or T1D mice, or 4 mg of SHP-1 inhibitor (SSG) was intradermally injected into the dorsal skin of WT C57BL/6 normal or T1D mice 18 and 6 h before wounding. Eight-millimetre excisional wounds were made using sterile biopsy punches. Wound areas were photographed every other day and calculated by Image J. Two-millimetre skin surrounding the wound edges or unwounded skin far from wounds was taken either for RNA isolation or for protein extraction or stored in formalin for haematoxylin and eosin staining.

**RegIIIγ neutralization in vivo.** After the mouse fur was removed, 100 μg of RegIIIγ-neutralizing antibody was intradermally injected into dorsal skin of WT C57BL/6 normal mice 18 and 6 h before wounding. Two-millimetre skin surrounding the wound edges or unwounded skin far from wounds was taken at indicated time points for western blot analysis.

**Cell culture and stimulation.** The NHEKs (Lifeline Cell Technology) were cultured in EpiLife medium (Invitrogen) containing EpiLife Defined Growth Supplement (EDGS, Invitrogen) and 60 mM $Ca^{2+}$. Adult human epidermal keratinocytes (AHEKs, SclenCell Research Laboratories) were cultured in keratinocyte medium (KM) following the manufacturer's introduction. Human embryonic kidney 293T cells gifted by Dr Jiemin Wong in East China Normal University were cultured in DMEM (Dulbecco's Modified Eagle's medium) with 10% fetal bovine serum (Gibco). All the cells were tested and were not contaminated by mycoplasma. For primary murine keratinocytes culture, the skin from newborn mice (2- or 3-day-old) was cut into small pieces and then suspended on dispase II (25 U ml$^{-1}$) overnight at 4 degree. Next day, the epidermis was peeled and placed into 5 ml of 0.05% trypsin (GIBCO) for 10 min at 37 degree. During incubation, the tube was shaken gently once. After 10 min, 10 ml cold TNS was added into 5 ml of trypsin and centrifuged at 500g for 10 min to make a pellet. Ten millimetres of Medium 154CF (Invitrogen) was added to resuspend the pellet. The cells were then seeded into a 24-well plate and cultured in Medium 154CF. The cells were grown to 80% confluence before the indicated doses of rhIL-33, rhIL-17, REG3A or different inhibitors were used to stimulate cells for indicated times. For glucose treatment, NHEKs or AKHEKs were pretreated with 20 mM glucose (D-glucose) or 20 mM mannitol for 15 min in the presence or absence of 2 mM advanced glycosylation end inhibitor aminoguanidine

(Santa Cruz Biotechnology) before stimulation with IL-17 (200 ng ml$^{-1}$). After 24 h, the cells were collected for protein extraction.

**Luciferase reporter assay.** The promoters of human TNF-α and IL-6 were amplified by polymerase chain reaction (PCR) and cloned into the pGL3-basic vector with a luciferase reporter (Promega). C-Jun binding sites on TNF-α and IL-6 promoter were predicted by ALGGEN and JASPAR websites. TNF-α promoter–reporter mutant lack of c-Jun binding site from bp-928 to bp-916 and IL-6 promoter–reporter mutant lack of two c-Jun bind sites (bp-1739 to bp-1733 and bp-135 to bp-129) were generated by PCR. The primers used were listed in Supplementary Table 2. Human embryonic kidney 293 T cells were seeded in 24-well plates one day before transfection with a mixture of pGL3 luciferase vector and different doses of the plasmid expressing c-Jun. Twenty-four hours post transfection, the cells were collected and luciferase activity was measured on LUMIstar OPTIMA microplate reader (BMG LRBTECH) by using the Dual-Luciferase Reporter Assay system (Promega). The ration of firefly luciferase to renilla luciferase was calculated for each well.

**shRNA/siRNA preparation and targeting gene knockdown.** For gene silencing, human SHP-1, IL-33 and REG3A specific shRNA (short hairpin RNA; Supplementary Table 3) were cloned into the pLL3.7 vector (Addgene) as the manufacturer described. Four micrograms of pLL3.7 constructs containing specific shRNAs, 4 mg of packaging plasmid psPAX2 (Addgene) and 2 μg of envelope plasmid pMD2.G (Addgene) were used to transfected human embryonic kidney 293T cells by calcium phosphate precipitation method. Forty-eight hours later, lentiviruses containing targeted gene shRNA were collected and used to transfect NHEK cells. Small interfering RNA (siRNA) oligonucleotides targeting EXTL3 and SHP-1 (Supplementary Table 4) were designed and synthesized by Genepharma (China). A mixture of two siRNA oligonucleotides for each EXTL3 or three siRNA oligonucleotides for each SHP-1 was used to transfected NHEKs. Blockage efficiency of shRNA or siRNA was tested by either western blot or PCR with reverse transcription (RT–PCR).

**Quantitative RT–PCR.** Total RNA was extracted from mouse wound homogenates or cells using TRIzol reagent (Life Technologies) following the manufacturer's introduction. Complementary DNA was synthesized using cDNA Reverse Transcriptase kit (Roche). Quantitative RT–PCR-specific primers are shown in Supplementary Table 5. Quantitative RT–PCR was performed in triplicate using SYBR green master mix (Roche) on a StepOnePlus Real-Time PCR System (Applied Biosystem). The samples with low yield of RNA were pre-determined and excluded.

**Enzyme-linked immunosorbent assay.** The supernatants of skin homogenate or cell culture supernatants were collected for cytokine evaluation. The samples with low yield of protein were pre-determined and excluded. Cytokine production was measured by ELISA kits of mTNF-α (BD Biosciences), hTNF-α (R&D systems), mIL-6 (R&D systems), hIL-6 (R&D systems), mIL-17 (R&D systems), mIL-33 (R&D systems) and RegIIIγ (Cloud Clone corp.) according to the manufacturer's instructions.

**Immunoprecipitation and immunoblot analysis.** For immunoprecipitation experiment, whole-cell extracts were prepared by using lysis buffer after transfection or stimulation and were incubated with the appropriated antibodies (listed in Supplementary Methods) overnight at 4 degree. Protein A&G beads (Abmart) were added and the incubation was continued for 4 h at 4 degree. The beads were then washed three times with washing buffer, and the immunoprecipitates were eluted with 2× sodium dodecyl sulfate (SDS) loading buffer and separated by PAGE (polyacrylamide gel electrophoresis). The proteins were transferred to nitrocellulose membrane and probed with the appropriate antibodies (listed in Supplementary Methods). Skin samples only for immunoblot analysis were homogenized in pre-cold lysis buffer containing protease inhibitor cocktail (Roche) by using Beadbeater (Biospec). Equal amount of total protein from each sample was separated with SDS–PAGE and then transferred to nitrocellulose membrane followed by probing with the indicated antibodies (listed in Supplementary Methods).

Uncropped images of the blots for Figs 1,2 and 4 are shown in Supplementary Fig. 10; uncropped images of the blots for Fig. 5 are shown in Supplementary Fig. 11; uncropped images of the blots for Fig. 6 are shown in Supplementary Fig. 12; uncropped images of the blots for Fig. 7 are shown in Supplementary Fig. 13; uncropped images of the blots for Supplementary Figs 2,3,5 and 6 are shown in Supplementary Fig. 14.

**Histological analysis and immunohistochemistry.** Five micrometres of formalin-fixed, paraffin-embedded tissue sections were mounted on glass slides. The sections were then dewaxed and subsequent pretreated with antigen retrieval solution. The sections were stained with anti-human REG3A antibody (Shanghai Immunogen Biological Technology &Co. LTD, A101122-LZ0098), or anti-mouse RegIIIγ antibody (ABclonal Biotechnology, WG-00077D-K39), or anti-human

IL-33 antibody (R&D, AF4810) or anti-mouse IL-33 antibody (R&D, AF3626), and then reprobed with the horseradish peroxidase-conjugated secondary antibody (Neobioscience, China) according to the manufacturer's introduction. For immunofluorescent staining of cells, the keratinocytes were fixed in 4% paraformaldehyde for 10 min followed by ice-cold 0.25% Triton X-100. The cells were stained with indicated antibodies (listed in Supplementary Information) and then nucleic acids were counterstained with DAPI before analysis with microscope. For all staining, inadequate staining samples due to technical problems were excluded.

**Statistical analysis.** All data are present as mean + s.e.m. Two-tailed $t$-tests were used to determine the significances between two groups. One-way or two-way analysis of variance with Bonferroni post test was used to analyse multiple groups. For all the statistical tests, $P$ values $< 0.05$ were considered to be statistically significant. All the relevant data are available from the authors.

**Data availability.** The data that support the findings of this study are available from the corresponding author upon reasonable request.

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

## Acknowledgements

We thank X. Lin from Tsinghua University for providing $Jnk1^{-/-}$ and $Jnk2^{-/-}$ mouse breeding pairs and C. Dong from Tsinghua University for $Il17^{-/-}$ mouse breeding pairs. This work was supported by National Key Research and Development Program of China (2016YFC0906200), National Natural Science Foundation of China (31222021, 31470878, 31170867 and 81202327), National Program for Support of Top-Notch Young Professionals to Y. Lai, the Science and Technology Commission of Shanghai Municipality (13JC1402301 and 11DZ2260300), Shanghai Education Commission (13SG25), New Century Excellent Talents in University (NCET-11-0141), and Henry Fok Educational Foundation (141017).

## Author contributions

Y.W., Y.Q., Y. Liu and Y. Lai designed the experiments and analysed the data; Y.W., Y.Q. and Y. Liu performed most of the experiments; Y.W. and K.L. did immunofluorescent staining; K.A.R. provided wounding samples from $Lepr^{db/db}$ mice and analysed RegIIIγ expression in these mice; D.L. and J.L. did co-immunoprecipitation of endogenous SHP-1 and JNK2; H. Lei performed the initiated experiments of the project; T.Z., H. Li, Z.W. and Z.J. evaluated IL-33 induced by IL-17; W.W. isolated primary murine keratinocytes; S.J. and Z.X. provided human samples; Y.W. and Y. Lai prepared the manuscript; Y. Lai wrote the manuscript.

**Additional information**

**Competing financial interests:** The authors declare no competing financial interests.

