## [Peer Review File · Nature Communications]

Reviewer #1 (Remarks to the Author):

Comments to the authors

In this manuscript entitled "Defective REG3A expression exacerbates skin inflammation in diabetes", Wu et al., reported a novel role of REG3A in controlling TLR3 mediated inflammatory signaling to maintain skin homeostasis after wound healing. The findings are very interesting and experiments were well performed, while several issues were raised by this reviewer.

General comments

1. The title of the manuscript impresses the readers with the relevance between REG3A and diabetic skin inflammation, while most part of the study deals with the connections of REG3A and TLR3, as well as their downstream signaling events. It seems that the upstream factors of REG3A, which are closer to the theme, appear not so highlighted.
2. Is REG3A highly expressed in immune cells? Have the authors checked the expression status of REG3A in immune cells after skin injury with or without diabetic influence?
3. As other toll-like receptors are also involved in the inflammatory responses during wound healing, is the regulatory effect of REG3A specific to TLR3?

Specific comments

Figure 1a: Why is epidermis of DM thicker than Nor?

Figure 1b: What does "Unw" or "W" stand for?

Figure 1f, g: If 100µg of RegIIIγ was intradermally injected into mouse dorsal skin, how much RegIIIβ can be physiologically generated in a certain amount of normal mouse wounds? Is it equivalent to 100µg?

Figure 1e: Size of lines used for SEM are not the same.

Figure 2:

1. It is not clear whether the authors should claim that "Hyperglycemia inhibits IL-17-induced IL-33 to decrease REG3A production", or better say "Hyperglycemia IL-33 to decrease REG3A production"?
2. It would be helpful to tell readers the rationality for why the authors investigated the relationship between IL-33 (but not other factors) and REG3A, are there any publications to support it?
3. Do the authors have any data to address how IL-33 induces REG3A expression in keratinocytes?
4. Page 7 line 2, it should be "Fig. 2j, k and Supplementary Fig. 2e", not "Fig. 3j, k and Supplementary Fig. 2e". Line 5, "Fig. 2k", not "Fig. 3k".

Figure 3:

1. What are expression levels of TLR-3 in normal skin and different kinds of wounds?
2. Since the authors showed that the poly (I:C) induced TNF- α and IL-6 via activating TLR-3, what signaling pathway(s) is involved in the inhibitory role of REG3A in TLR3-mediated pro-inflammatory cytokines such as TNF- α and IL-6?
3. What about wound healing in TLR-3^{-/-} compared with WT mice?

Figure 4:

In Fig.4h, poly (I:C) failed to induce more TNF- α when SHP-1 was knocked-down compared to the scramble group. This paradox should be clearly addressed.

Figure 5 + Figure 6:

1. The authors claim that REG3A induces SHP-1 expression through the EXTL3-PI3K-AKT-STAT3 signaling pathway, and inhibits TLR3-activated JNK2 in keratinocytes. Expression levels and/or activity of AP-1, a direct downstream molecule of JNK1/2 should be investigated in normal skin and different kinds of wounds.
2. Does activation of JNK1/2-AP1 signaling directly induce expression of TNF- α and IL-6 or indirectly? Because the authors found that activating TLR-3 triggered expression of TNF- α and IL-6 which are target genes induced by activation of NF- κ B signaling.

Finally, if the REG3A is an antimicrobial protein in keratinocytes, why it should inhibit the activation of TLR-3 that is physiologically required for clearing viruses? This should be discussed in the section of DISCUSSION.

Reviewer #2 (Remarks to the Author):

Wu, et al. describe a role for REG3A plays in controlling inflammation in diabetic wounds. They describe a novel pathway in diabetes wounds in which IL-33 leads to production of REG3A which subsequently induces SHP1 that ultimate inhibits TLR3-induced inflammation during wound healing. Although the pathway is novel and the results are interesting, there is an abundance of data that does not ultimately contribute to the manuscript and makes the results difficult to interpret (the overabundance of data should be removed from the manuscript). In addition, almost all of the in vitro work essentially uses neonatal human keratinocytes without any tests in adult primary human keratinocytes or in more sophisticated human skin equivalent cultures. As the neonatal cells have a higher plating efficiency and regeneration capacity than adult cells and the interpretations are thus limited and adult keratinocytes need to be used to confirm results. Immunoblots are also used throughout without confirmatory more quantitative data. There is a selective use of in vivo data with key in vivo experiments missing (e.g. TLR3^{-/-}). Also, the results with JNK2 are not convincing. Finally, throughout the manuscript, the statistical analyses need to be better described along with group numbers that are appropriate.

Major Revisions

Fig. 1. The authors show that REG3A and RegIIIgamma are decreased in diabetic wounds and administration of RegIIIgamma to diabetic wounds promotes decreased proinflammatory cytokine (TNFalpha and IL-6) and enhanced wound healing. Fig. 1a. Please show low and high magnification of the skin from normal and diabetic skin so these differences could be better seen. Furthermore, to confirm the results in Fig. 1a quantifying the REGA3 expression by western or QPCR. Fig. 1i, the representative images do not match the size of the wounds in the data presented in Fig. 1h. The authors also claim that application of RegIIIgamma correlates with the degree of inflammation and leukocyte influx as they show histology in Supplemental Fig. 1. However, to make this claim the numbers of leukocytes in the tissue sections should be enumerated by manual counting or FACS analysis.

Fig. 2. The data show that IL-33 is decreased in diabetic wounds and the IL-33 mediated induction of REG3A is induced by IL-17 in nondiabetic conditions but not in the setting of hyperglycemia or diabetes and IL-33 administration enhances wound healing in normal and diabetic mice. Please provide information into why IL-33 was evaluated. As this is an IL-1 cytokine family member, do other IL-1 family members (IL-1alpha/beta, IL-18, IL-36alpha,beta,gamma) or TLRs regulate REG3A in a similar manner? Fig. 2a, b and c, the findings with the immunoblots and representative histology sections for IL-33 needs be quantified by ELISA to confirm results. In Fig. 2b and c, why are there 2 images for each group? High and low magnification images should be provided.

Fig. 3. In this figure, it would be helpful to indicated that all of the "WT" and "TLR3-/-" mice used are diabetics and a similar labeling system with "DM" or "DM TLR3-/-" would be more clear. How many mice are in Fig. 3C?-and please include error bars. Furthermore, the in vivo healing rate data for the normal and diabetic TLR3-/- mice to correlate with the findings in Fig. 1h and 2m.

Fig. 4. The authors demonstrate that REG3A and RegIIIgamma induce SHP1 and SHP1 inhibits TLR3 induced TNFalpha. The data in Fig. 4e are subtle and need to be confirmed with ELISA or QPCR. Additionally, Fig.4i does not show statistically significant differences of TNF-alpha expression in DM treated mice although this is interpreted in the text. The interpretation must be changed or the experiment repeated to show increased TNF-alpha expression upon treatment. It is also unclear why the authors chose to continue experiments only with TNFalpha when they had significant prior data with IL-6. For completeness, the IL-6 data should be included. In addition, the data with the SSG inhibitor in Fig. 7h should be relocated to this figure as it relates directly to this data with SHP1. In addition, please show statistics on Fig. 7h between each of the compared groups.

Fig. 6. Overall the data involving JNK is tenuous at best. Fig. 6a the slight statistical decrease in TNFalpha levels is likely not biologically relevant. In Fig. 6b, p-JNK levels are not very different with the inhibitor. In Fig. 6d, the p-JNK2 data all have similar levels of expression. In Fig. 6i, the slightly over 2 fold increase in TNFalpha gene expression is subtle. In Fig. 6j, why are the time points different in the graph compared to Fig. 1h or Fig. 2m. Also the in vivo findings are subtle.

Fig. 7. The data in Fig. 7g are difficult to interpret because the SHP-1 si treatment resulted in 20-30% decrease in healing rate percentage compared to the scramble treatment. This makes the ability of the SHP-1 si to inhibit the Poly(I:C) + REG3A group difficult to interpret because the healing rate is not as high as the scramble treatment. Please see above Fig. 4 regarding Fig. 7h.

Finally, Figure including a model of how this system works would be helpful to interpret the results.

Minor Revisions

Pg. 5 - change BABL/c should be Balb/c

Pg. 15 line 3 - Change TLR3-activatd to TLR3-activated.

Pg. 19 line 10 - Provide list of antibodies used for Immunoprecipitation and Immunoblot analysis.

Reviewer #3 (Remarks to the Author):

Wu et al., Defective REG3A

Comments and questions:

The writing has a few mistakes in the English, which need to be tidied up. The study contains a huge amount of work and very interesting results but a lot of it is very disparate with inconsistent animal models being used. For instance, it is odd that 3 different animal models of diabetes have been used and it is not clear why. The STZ model (the only FDA approved model of diabetic wound healing) is sufficient? Why were two different strains of mouse used? In the text and figures it is rarely clarified what kind of diabetic mouse was used and this needs to be clarified.

Many researchers in diabetes would not regard the db/db 11 mM and NOD 7mM Glucose levels to actually be diabetic and would want to see levels over 20 mM as was seen with the STZ mice. The methods suggest that blood glucose over 15 mM was taken as diabetic but these are clearly not. This leads me to question if the db/db and NOD mice were really diabetic - this then draws into question the data from these mice.

The presentation of histological data and staining is problematic. Whilst all of the figure legends describe wound edge, a wound edge is never shown! You really need to be able to see the wound edge and changes that take place at the wound edge and you need to be able to see them declining away from the wound edge. All of the images that are presented could be anywhere in intact skin. Also it is important to consider what is happening in the dermis as this plays a very

important role (probably greater than the epidermis) in inflammation and wound healing. Why has this been ignored in this study? As it stands the way the data has been presented I am unable to properly review the results of the histology.

General points that need addressing/clarification:

Fig 1 A You cannot see the wound edge in these images and this is required to be able to tell that it is wound and not normal tissue.

What type of wound is this and how old is it?

Fig 1 I - are these images the median values from the data set or the best/worst? How many mice were used in the wound healing studies - it is not clear.

Sup 1. b, c & d wound edge images - we really need to be able to see the wound edge and we can't. This could be anywhere.

Why have the different mag between b & c?

b seems to show some staining in the dermis - what structures is this in? Blood vessels? It is not clear from the pictures but this could also be interesting and is not addressed.

Fig 2 a The legend describes this as day 3 wound but the figure appears to be a time course, 0, 1 3, 5 & 7.

I find this result rather odd that REG3 turns on upon wounding and then remains constant for 7 days in normal wounds. I would have expected changes in line with increasing and then decreasing as inflammation resolves.

Fig 2b is supposed to be a skin wound but I can see no details of the tissue to tell what cell types these are or where they are in relation to the wound.

P 7 high glucose experiment. There are no experimental details given in the methods for this. How long were the cells cultured for as it can take some time (several days) for the glucose to take effect. This experiment could really do with a Mannitol control for osmolarity. Line 2 & 5 Fig 3k should be 2k. Fig 2 e is the scale bar correct at 25um or are the nuclei huge?

Fig 2 L - what type of diabetic mice are these? Needs to be clarified as 3 types are being used. Same goes for Fig 3 and Fig 4 I

Fig 4 f. I am unable to see the wound edge in these images. I am unable to see anything in the db/db images.

Point-by-Point Response

Reviewer #1:

In this manuscript entitled "Defective REG3A expression exacerbates skin inflammation in diabetes", Wu et al., reported a novel role of REG3A in controlling TLR3 mediated inflammatory signaling to maintain skin homeostasis after wound healing. The findings are very interesting and experiments were well performed, while several issues were raised by this reviewer.

General comments

Comment 1. The title of the manuscript impresses the readers with the relevance between REG3A and diabetic skin inflammation, while most part of the study deals with the connections of REG3A and TLR3, as well as their downstream signaling events. It seems that the upstream factors of REG3A, which are closer to the theme, appear not so highlighted.

Response: As the reviewer suggested, we changed our title to "Hyperglycemia inhibits REG3A expression to exacerbate TLR3-mediated skin inflammation in diabetes".

Comment 2. Is REG3A highly expressed in immune cells? Have the authors checked the expression status of REG3A in immune cells after skin injury with or without diabetic influence?

Response: Accumulating evidence has shown that REG3A/RegIII γ is highly expressed in epithelial cells, paneth cells, enterocytes, β cells and cancer cell lines, but not in immune cells¹⁻⁷. Moreover, we have reported that REG3A/RegIII γ was highly expressed in epidermal keratinocytes but not in other cells after skin injury⁵. Furthermore, our immunohistochemical staining data showed that in diabetic condition RegIII γ was primarily expressed in epidermal keratinocytes in skin wounds of *Lepr^{db/+}* mice compared to *Lepr^{db/db}*, but not in dermis where leukocytes were recruited (Figure 1 attached to this point-by-point response). Thereby we did not further check the expression status of REG3A in immune cells after skin injury with or without diabetic influence.

Comment 3. As other toll-like receptors are also involved in the inflammatory responses during wound healing, is the regulatory effect of REG3A specific to TLR3?

Response: Yes, other toll-like receptors such as TLR4 have been shown to regulate inflammatory responses during wound healing⁸. However, the regulatory effect of REG3A is specific to TLR3 in keratinocytes as we did not observe that TLR4 ligand, LPS, induced inflammatory response in keratinocytes⁹, and REG3A did not inhibit poly(I:C)-induced inflammatory response in THP-1 cells (Supplementary Figure 4d & e).

Specific comments

Comment Figure 1a: Why is epidermis of DM thicker than Nor?

Response: The samples (last version) we detected were from chronic wounds in diabetic patients. The observation of thick epidermis in chronic wounds from diabetic patients is consistent with the previous report that chronic wounds from diabetic patients show hyperproliferative and nonmigratory epidermis, and unresolved inflammation¹⁰. In this revised version, we re-collected samples from normal patients with acute injury and diabetic patients with acute injury, and re-analyzed REG3A production in wounded skin by IHC. From Figure 1b we can see that the production of REG3A was markedly decreased in skin wounds of diabetic patients and the epidermis of diabetic patients was not thicker than that of normal patients.

Comment Figure 1b: What does "Unw" or "W" stand for?

Response: "Unw" stands for "Unwounded skin", "W" stands for "Wounded skin". We have added the information in Figure 1 legend.

Comment Figure 1f, g: If 100µg of RegIIIγ was intradermally injected into mouse dorsal skin, how much RegIIIγ can be physiologically generated in a certain amount of normal mouse wounds? Is it equivalent to 100µg?

Response: The concentration of RegIIIγ in normal skin wounds was around 1.5ng/mg by ELISA (Figure 2c). Usually, we consider 2mm skin (100-200mg) surrounding the wound edge as wounded skin, and the physiological amount of RegIIIγ in normal skin wounds was 150-300ng. We thereby checked whether RegIIIγ could promote wound healing in diabetic mice in a dose-dependent manner and found that only 1µg RegIIIγ significantly promoted wound healing in diabetic mice compared to controls (See Figure 2a attached to this point-by-point response). Specifically, 100µg RegIIIγ promoted wound healing most potently (Figure 2b attached to this point-by-point response, also see Figure 1h in the main text).

Comment Figure 1e: Size of lines used for SEM are not the same.

Response: Thanks for the reviewer's kindly reminder. We have changed the error bars to the same size.

Figure 2:

Comment 1. It is not clear whether the authors should claim that "Hyperglycemia inhibits IL-17-induced IL-33 to decrease REG3A production", or better say "Hyperglycemia IL-33 to decrease REG3A production"?

Response: As the reviewer suggested, we changed "Hyperglycemia inhibits IL-17-induced IL-33 to decrease REG3A production" to "Hyperglycemia inhibits IL-33 to decrease REG3A production".

Comment 2. It would be helpful to tell readers the rationality for why the authors

investigated the relationship between IL-33 (but not other factors) and REG3A, are there any publications to support it?

Response: As we described in Paragraph 2, page 6, we first checked the expression of IL-17 that has been shown to induce REG3A expression in keratinocytes⁵, but we did not observe that IL-17 production was decreased as similar as RegIII γ in skin wounds of diabetic mice (Figure. 2a and Supplemental Figure 2a). In addition to IL-17, we screened multiple cytokines including TNF- α , IL-6, IFN- γ , IL-13, IL-12 and IL-22 that are elevated in skin wounds, but these cytokines did not induce REG3A expression in keratinocytes⁵. We then further screened other cytokines or TLR ligands that might influence REG3A expression in keratinocytes. We found that recombinant human IL-33(rhIL-33) and rhIL-36 γ , but not other cytokines or TLR ligands, induced REG3A expression in keratinocytes (Supplementary Fig. 2b). Therefore, we hypothesized that the reduction of REG3A might be due to decreased IL-33 or IL-36 γ in skin wounds of diabetes. The *in vivo* experiments showed that IL-33, but not IL-36 γ , was decreased in a similar expression pattern of RegIII γ in skin wounds of T1D mice as well as patients with diabetes (Fig. 2c-e and Supplementary Fig. 2a), which was consistent with the previous reported that IL-33 was decreased in diabetic skin wounds¹¹. We thereby assumed that the reduction of RegIII γ could be due to decreased IL-33 in skin wounds of diabetes and further confirmed it.

Comment 3. Do the authors have any data to address how IL-33 induces REG3A expression in keratinocytes?

Response: Yes, we have observed that IL-33 activates ST2-AKT-STAT3- β -catenin signaling pathway to induce REG3A expression in keratinocytes (See Figure 3 attached to this point-by-point response). To maintain clarity and brevity of the original report, we did not include these data in the main figures or the supplementary figures and only showed them in this point-by-point response. Of course, should the reviewer wish us to include these data in the main body of the manuscript, we would be happy to restructure as you see fit.

Comment 4. Page 7 line 2, it should be "Fig. 2j, k and Supplementary Fig. 2e", not "Fig. 3j, k and Supplementary Fig. 2e". Line 5, "Fig. 2k", not "Fig. 3k".

Response: Sorry for typos. We have corrected them.

Figure 3:

Comment 1. What are expression levels of TLR-3 in normal skin and different kinds of wounds?

Response: We checked the expression of TLR3 in normal skin and different kinds of wounds from mice by using RT-PCR and found that TLR3 expression was increased in skin wounds of both normal and diabetic mice (Figure 3a). Particularly, TLR3 expression in normal and wounded skin of diabetic mice was higher than that in normal mice (Figure 3a).

Comment 2. Since the authors showed that the poly (I:C) induced TNF- α and IL-6 via activating TLR-3, what signaling pathway(s) is involved in the inhibitory role of REG3A in TLR3-mediated pro-inflammatory cytokines such as TNF- α and IL-6?

Response: In our study, we found that IL-33 induced REG3A expression and that REG3A exerted its anti-inflammatory activity on TLR3-mediated inflammatory response via the induction of SHP-1 in skin wounds. However, under diabetic condition, hyperglycemia inhibited the production of IL-33, thus resulting in decreased REG3A and REG3A-induced SHP-1 but increased production of TLR3-mediated pro-inflammatory cytokines (Figure 7h).

Comment 3. What about wound healing in TLR-3^{-/-} compared with WT mice?

Response: Wound healing in STZ-induced *Tlr3*^{-/-} diabetic mice was significantly increased compared to STZ-induced wild-type diabetic mice, although the deficiency of *Tlr3* in normal mice had minimal effect on delaying wound healing compared to wild-type normal mice (Figure 3c). The data suggests that under normal condition the appropriate inflammatory response is required for normal wound healing, while in diabetes the activation of TLR3 led to the excessive production of pro-inflammatory cytokines such as TNF- α and IL-6, thus resulting in delayed wound healing.

Figure 4:

Comment In Fig.4h, poly (I:C) failed to induce more TNF- α when SHP-1 was knocked-down compared to the scramble group. This paradox should be clearly addressed.

Response: It is probably a misunderstood. Our data showed that poly(I:C) significantly induced TNF- α expression after SHP-1 was knocked down (Figure 4h, black bar in the right), but REG3A failed to inhibit poly(I:C)-induced TNF- α expression after SHP-1 was knocked down(Figure 4h, grey bar in the right).

Figure 5 + Figure 6:

Comment 1. The authors claim that REG3A induces SHP-1 expression through the EXTL3-PI3K-AKT-STAT3 signaling pathway, and inhibits TLR3-activated JNK2 in keratinocytes. Expression levels and/or activity of AP-1, a direct downstream molecule of JNK1/2 should be investigated in normal skin and different kinds of wounds.

Response: AP-1 is a complex composed of c-Fos, c-Jun and ATF protein families. JNKs were originally identified as kinases that bind and phosphorylate c-Jun on Ser-63 and Ser-73 within its transcriptional activation domain. We thereby tested whether the activation of JNK would lead to the activation of c-Jun. Western blot analyses showed that poly(I:C) induced the phosphorylation of JNK and c-Jun and the inhibition of JNK by its inhibitor decreased the phosphorylation of c-Jun in primary human keratinocytes (Figure 6c). Moreover, poly(I:C) induced the phosphorylation of

JNK2 and c-Jun in primary wild-type murine keratinocytes while the deficiency of *Jnk2* not *Jnk1* completely blocked c-Jun phosphorylation induced by poly(I:C) (Figure 6f-h). All these data confirm that c-Jun, one of the components of AP-1, is downstream molecule of JNK2.

Comment 2. *Does activation of JNK1/2-AP1 signaling directly induce expression of TNF- α and IL-6 or indirectly? Because the authors found that activating TLR-3 triggered expression of TNF- α and IL-6 which are target genes induced by activation of NF- κ B signaling.*

Response: We analyzed the promoters of TNF- α and IL-6 and found that both promoters have c-Jun binding site. Thereby we did the luciferase assay to test whether the overexpression of c-Jun would activate TNF- α promoter or IL-6 promoter. From Figure 6d,e we can see that c-Jun activated TNF- α promoter or IL-6 promoter in a dose-dependent manner while this induction was abrogated or attenuated when c-Jun binding site was mutated in TNF- α promoter or IL-6 promoter, suggesting that c-Jun, one of the components of AP-1, directly induces the expression of TNF- α and IL-6.

Comment 3. *Finally, if the REG3A is an antimicrobial protein in keratinocytes, why it should inhibit the activation of TLR-3 that is physiologically required for clearing viruses? This should be discussed in the section of DISCUSSION.*

Response: REG3A only inhibited TLR3-induced proinflammatory cytokines such as TNF- α and IL-6 (Figure 3d-g), but not TLR3-induced IFN β (Supplementary Figure 4c) that is physiologically required for clearing viruses, suggesting REG3A plays a critical role in balancing inflammatory responses in skin wounds as well as keeping its important role in clearing viruses. All these have been added in the section of Discussion.

Reviewer #2 (Remarks to the Author):

Comment: *Wu, et al. describe a role for REG3A plays in controlling inflammation in diabetic wounds. They describe a novel pathway in diabetes wounds in which IL-33 leads to production of REG3A which subsequently induces SHP1 that ultimate inhibits TLR3-induced inflammation during wound healing. Although the pathway is novel and the results are interesting, there is an abundance of data that does not ultimately contribute to the manuscript and makes the results difficult to interpret (the overabundance of data should be removed from the manuscript). In addition, almost all of the in vitro work essentially uses neonatal human keratinocytes without any tests in adult primary human keratinocytes or in more sophisticated human skin equivalent cultures. As the neonatal cells have a higher plating efficiency and regeneration capacity than adult cells and the interpretations are thus limited and adult keratinocytes need to be used to confirm results. Immunoblots are also used throughout without confirmatory more quantitative data. There is a selective use of in vivo data with key in vivo experiments missing (e.g. TLR3^{-/-}). Also, the results with*

JNK2 are not convincing. Finally, throughout the manuscript, the statistical analyses need to be better described along with group numbers that are appropriate.

Response: As the reviewer suggested, we conducted a series of experiments in adult human keratinocytes to confirm the results from neonatal human keratinocytes, including the induction of IL-33 by IL-17 and the inhibition of IL-33 by high glucose (Supplementary Figure 2d, 3b), the inhibitory effect of REG3A and c-REG3A on poly(I:C)-induced TNF- α and IL-6 (Figure 3g and Supplementary Figure 4g), the induction of SHP-1 by REG3A (Figure 4b), the inhibitory effect of REG3A on poly(I:C)-induced TNF- α and IL-6 before and after SHP-1 was silenced (Figure 4i), the activation of JNK by poly(I:C) to induce TNF- α and IL-6 (Figure 6b), and the inhibitory effect of REG3A on poly(I:C)-induced JNK2 phosphorylation (Figure 6o). We also examined the healing process in wild type versus *Tlr3*^{-/-} normal and diabetic mice and showed that wound healing in STZ-induced *Tlr3*^{-/-} diabetic mice was significantly increased compared to STZ-induced wild-type diabetic mice (Figure 3c). Furthermore, we redid the western blot of p-JNK induced by poly(I:C) after JNK was inhibited (Figure 6c), re-evaluated the production of TNF- α and IL-6 in WT and *Jnk2*^{-/-} mice instead of JNK inhibitor in vivo experiment (Figure 6k), re-tested the healing process in WT, *Jnk1*^{-/-} and *Jnk2*^{-/-} normal and diabetic mice (Figure 6l and Supplementary Figure 7a), deleted some unnecessary data, used image J to analyze the densitometry of all the bands from western blots, and added the group numbers along with the statistical analyses.

Major Revisions

Comment Fig. 1. The authors show that REG3A and RegIIIgamma are decreased in diabetic wounds and administration of RegIIIgamma to diabetic wounds promotes decreased proinflammatory cytokine (TNFalpha and IL-6) and enhanced wound healing. Fig. 1a. Please show low and high magnification of the skin from normal and diabetic skin so these differences could be better seen. Furthermore, to confirm the results in Fig. 1a quantifying the REGA3 expression by western or QPCR. Fig. 1i, the representative images do not match the size of the wounds in the data presented in Fig. 1h. The authors also claim that application of RegIIIgamma correlates with the degree of inflammation and leukocyte influx as they show histology in Supplemental Fig. 1. However, to make this claim the numbers of leukocytes in the tissue sections should be enumerated by manual counting or FACS analysis.

Response: As the reviewer suggested, we showed the low and high magnifications of the skin from normal and diabetic skin in Figure 1b in the revised version, analyzed the mRNA expression of REG3A by RT-PCR in skin wounds of normal and diabetic patients (Figure 1a).

Moreover, we double-checked all photos of wounds and chose the representative images that match the size of the wounds in the data presented in Figure 1h.

Furthermore, we manually counted leukocytes in the tissue sections and added the data of leukocyte number in Supplementary Figure 1d.

Comment Fig. 2. The data show that IL-33 is decreased in diabetic wounds and the IL-33 mediated induction of REG3A is induced by IL-17 in nondiabetic conditions but not in the setting of hyperglycemia or diabetes and IL-33 administration enhances wound healing in normal and diabetic mice. Please provide information into why IL-33 was evaluated. As this is an IL-1 cytokine family member, do other IL-1 family members (IL-1alpha/beta, IL-18, IL-36alpha, beta, gamma) or TLRs regulate REG3A in a similar manner? Fig. 2a, b and c, the findings with the immunoblots and representative histology sections for IL-33 needs be quantified by ELISA to confirm results. In Fig. 2b and c, why are there 2 images for each group? High and low magnification images should be provided.

Response: As we described in Paragraph 2, page 6, we first checked the expression of IL-17 that has been shown to induce REG3A expression in keratinocytes⁵, but we did not observe that IL-17 production was decreased as similar as RegIII γ in skin wounds of diabetic mice (Figure. 2a and Supplementary Figure 2a). In our previous studies we screened multiple cytokines including TNF- α , IL-6, IFN- γ , IL-13, IL-12 and IL-22 that are elevated in skin wounds, but we did not observe that these cytokines induced REG3A expression in keratinocytes⁵.

We have also tested whether other IL-1 family members (IL-1a, IL-18, IL-36 γ) or TLR ligands (Imiquimod, LTA, LPS) would regulate REG3A expression. We found that recombinant human IL-33(rhIL-33) and rhIL-36 γ , but not other cytokines or TLR ligands, induced REG3A expression in keratinocytes (Supplementary Fig. 2b). Therefore, we hypothesized that the reduction of REG3A might be due to decreased IL-33 or IL-36 γ in skin wounds of diabetes. The *in vivo* experiments showed that IL-33, but not IL-36 γ , was decreased in a similar expression pattern of RegIII γ in skin wounds of T1D mice as well as patients with diabetes (Fig. 2c-e and Supplementary Fig. 2a). All these data suggest that decreased REG3A/RegIII γ is due to decreased IL-33 in skin wounds of diabetes. We have added this information in page 6, paragraph 2.

For Fig. 2a, b and c, we did the ELISA to quantify IL-17, RegIII γ and IL-33 and showed that the production of RegIII γ and IL-33, but not IL-17, was significantly decreased in skin wounds of diabetic mice compared to those in normal mice (Figure 2a,c). Histology sections in Figure 2d were day-3 skin wounds in Figure 2c. ELISA data showed that the production of IL-33 in day-3 skin wounds of diabetic mice was significantly decreased compared to that in day-3 skin wounds of normal mice (Figure 2c). Histology sections in Figure 2e were human samples. Although ELISA assay was not available due to limited samples, IHC data showed that the production of IL-33 in skin wounds of diabetic patients was markedly decreased compared to that in normal patients (Figure 2e). Moreover, we added one more sample in Figure 2d, 2e and showed high and low magnifications in these two figures.

Comment Fig. 3. In this figure, it would be helpful to indicated that all of the "WT" and "TLR3-/-" mice used are diabetics and a similar labeling system with "DM" or "DM TLR3-/-" would be more clear. How many mice are in Fig. 3C?-and please

include error bars. Furthermore, the *in vivo* healing rate data for the normal and diabetic TLR3^{-/-} mice to correlate with the findings in Fig. 1h and 2m.

Response: As the reviewer suggested, we changed “WT” to “Nor WT” or “DM WT” and “Tlr3^{-/-}” to “Nor Tlr3^{-/-}” or “DM Tlr3^{-/-}” in Figure 3b-d.

In Figure 3c (Now Figure 3e in the revised version), we tested the inhibitory effect of REG3A on TLR3 ligand poly(I:C)-induced TNF- α and IL-6 in primary human keratinocytes, not in mice. Actually, there were error bars in Figure 3e. It's hard to see error bars because the data were reproducible well.

We also examined the healing process in wild type versus Tlr3^{-/-} normal and diabetic mice, and showed that wound healing in STZ-induced Tlr3^{-/-} diabetic mice was significantly increased compared to STZ-induced wild-type diabetic mice, although the deficiency of Tlr3 had a minimal effect on delaying wound healing compared to wild-type normal mice (Figure 3c). The data suggests that under normal condition the appropriate inflammatory response is required for normal wound healing, while in diabetes the activation of TLR3 led to the excessive production of pro-inflammatory cytokines such as TNF- α and IL-6, thus resulting in delayed wound healing.

Comment Fig. 4. The authors demonstrate that REG3A and RegIII γ induce SHP1 and SHP1 inhibits TLR3 induced TNF α . The data in Fig. 4e are subtle and need to be confirmed with ELISA or QPCR. Additionally, Fig.4i does not show statistically significant differences of TNF-alpha expression in DM treated mice although this is interpreted in the text. The interpretation must be changed or the experiment repeated to show increased TNF-alpha expression upon treatment. It is also unclear why the authors chose to continue experiments only with TNFalpha when they had significant prior data with IL-6. For completeness, the IL-6 data should be included. In addition, the data with the SSG inhibitor in Fig. 7h should be relocated to this figure as it relates directly to this data with SHP1. In addition, please show statistics on Fig. 7h between each of the compared groups.

Response: For Fig. 4e (Fig. 4f in the revised version), we shortened the exposure time and showed that rhIL-33-induced SHP-1 was markedly decreased after REG3A was knocked down. We also used Image J to analyze the densitometry of the bands and showed that the densities of SHP-1 bands were markedly decreased, which was consistent with the data from western blot (Figure 4f). Moreover, we have showed that rhIL-33-induced SHP-1 mRNA was markedly decreased after REG3A was knocked down and also used Image J to confirm it (Supplementary Figure 5e).

For Fig. 4i (Fig. 4j in the revised version), we redid the mouse experiment and reevaluated the expression of TNF- α and IL-6 expression in skin wounds before and after SSG treatment. We showed that the mRNA expression of TNF- α and IL-6 was significantly increased in skin wound of normal and diabetic mice after SSG treatment (Figure 4j). Moreover, we reevaluated IL-6 expression in Fig. 4g and 4h and added the data of IL-6 expression in Fig. 4g, 4h.

For Figure 7h (Fig.7g in the revised version), it showed that SHP-1 inhibitor SSG significantly inhibited the healing process that was restored by RegIII γ in

diabetic, but not only compared the healing process between normal and diabetic mice before and after SHP-1 was inhibited by SSG. We thereby kept Figure 7g in Figure 7. Meanwhile, we added statistics on Fig. 7g between each of the compared groups as the reviewer suggested.

Comment Fig. 6. Overall the data involving JNK is tenuous at best. Fig. 6a the slight statistical decrease in TNF α levels is likely not biologically relevant. In Fig. 6b, p-JNK levels are not very different with the inhibitor. In Fig. 6d, the p-JNK2 data all have similar levels of expression. In Fig. 6i, the slightly over 2 fold increase in TNF α gene expression is subtle. In Fig. 6j, why are the time points different in the graph compared to Fig. 1h or Fig. 2m. Also the *in vivo* findings are subtle.

Response: We redid all the experiments critiqued by the reviewer and added the new data in Figure 6. Figure 6a,b showed that JNK inhibitor sp600125 significantly inhibited the production of TNF- α and IL-6 in both primary and adult human keratinocytes. Figure 6c showed that JNK inhibitor markedly inhibited poly(I:C)-induced JNK phosphorylation in primary human keratinocytes.

For Figure 6d (now Figure 6g in the revised version), we redid western blot and showed that poly(I:C) induced JNK2 phosphorylation in a time-dependent manner with the maximum induction observed at 0.75h poly(I:C) stimulation. Consistent with this, the phosphorylation of c-Jun, a downstream molecule of JNK2, was induced by poly(I:C) in a time-dependent manner.

For Figure 6i (now Figure 6k in the revised version), we deleted the data of SSG inhibition *in vivo* and tested the production of TNF- α and IL-6 in skin wounds of WT and *Jnk2*-deficient normal and diabetic mice by ELISA, and found that both of cytokines were significantly decreased in skin wounds of *Jnk2*-deficient normal and diabetic mice compared to controls. Moreover, we redid the wound healing of WT, *Jnk1*^{-/-} and *Jnk2*^{-/-} diabetic mice and showed the healing rates at day 1, 3, 5, 7, 9 as what we did in Figure 1h and Figure 2g. Figure 6l showed that wound healing in *Jnk2*^{-/-} diabetic mice was significantly increased compared to wound healing in WT and *Jnk1*^{-/-} diabetic mice.

Comment Fig. 7. The data in Fig. 7g are difficult to interpret because the SHP-1 si treatment resulted in 20-30% decrease in healing rate percentage compared to the scramble treatment. This makes the ability of the SHP-1 si to inhibit the Poly(I:C) + REG3A group difficult to interpret because the healing rate is not as high as the scramble treatment. Please see above Fig. 4 regarding Fig. 7h.

Response: As requested, we deleted Figure 7g. However, we kept Figure 7h (now Figure 7g in the revised version) in Figure 7 because Figure 7h showed that SHP-1 inhibitor SSG significantly inhibited the healing process that was restored by RegIII γ in diabetic, but not only compared the healing process between normal and diabetic mice before and after SHP-1 was inhibited by SSG. Please also see response to Comment on Figure 4.

Comment Finally, Figure including a model of how this system works would be

helpful to interpret the results.

Response: As suggested, we added the schematic graph in Figure 7h to reflect how this system works.

Minor Revisions

Comment Pg. 5 - change BABL/c should be Balb/c

Response: We have corrected the typo.

Comment Pg. 15 line 3 - Change TLR3-activatd to TLR3-activated.

Response: We changed “TLR3-activatd” to “TLR3-activated”.

Comment Pg. 19 line 10 - Provide list of antibodies used for Immunoprecipitation and Immunoblot analysis.

Response: As the reviewer suggested, we added a list of antibodies used for immunoprecipitation and immunoblot analysis in Supplementary information.

Reviewer #3 (Remarks to the Author):

Comments and questions:

Comment: The writing has a few mistakes in the English, which need to be tidied up. The study contains a huge amount of work and very interesting results but a lot of it is very disparate with inconsistent animal models being used. For instance, it is odd that 3 different animal models of diabetes have been used and it is not clear why. The STZ model (the only FDA approved model of diabetic wound healing) is sufficient? Why were two different strains of mouse used? In the text and figures it is rarely clarified what kind of diabetic mouse was used and this needs to be clarified.

Response: We apologize that the mistakes in English. We have corrected all grammar errors and typos in the revised version.

The reason why we used 3 different animal models of diabetes was that we tried to show that the function of REG3A in the regulation of inflammatory responses in skin wounds was common in different types of diabetes. Since the reviewer thought that STZ model was sufficient, we removed the data of NOD and db/db mice from the main figures and relocated them in Supplementary Figure 1. The reason why we used both C57BL/6 and BAL/c mice was to show that REG3A regulated the inflammatory response in skin wounds of different mouse strains. In Figure 1 legend we clarified that all mice used in the study without specific notes were C57BL/6 strain, and T1D mice were developed by STZ induction.

Comment: Many researchers in diabetes would not regard the db/db 11 mM and NOD 7mM Glucose levels to actually be diabetic and would want to see levels over 20 mM as was seen with the STZ mice. The methods suggest that blood glucose over 15 mM

was taken as diabetic but these are clearly not. This leads me to question if the db/db and NOD mice were really diabetic - this then draws into question the data from these mice.

Response: That the blood glucose over 15 mM was considered as diabetic was referred to the previous report by Taniguchi K¹². We re-tested the concentration of blood glucose and chose mice with blood glucose over 20 mM as diabetic mice (Supplementary Figure 3a and Figure 4 attached to this point-by-point response), and then re-tested the expression of RegIII γ in skin wounds of these mice (Supplementary Figure 1b,c). We also corrected this in Methods section.

Comment: The presentation of histological data and staining is problematic. Whilst all of the figure legends describe wound edge, a wound edge is never shown! You really need to be able to see the wound edge and changes that take place are at the wound edge and you need to be able to see them declining away from the wound edge. All of the images that are presented could be anywhere in intact skin. Also it is important to consider what is happening in the dermis as this plays a very important role (probably greater than the epidermis) in inflammation and wound healing. Why has this been ignored in this study? As it stands the way the data has been presented I am unable to properly review the results of the histology.

Response: Overall we believe these critiques have now been resolved as we redid all the histological analyses and showed the wound edges in all sections (Figure 1b,e and Figure 2d,e)

The reason that we did not consider what is happening in the dermis is that our previous publication has showed that REG3A was highly expressed in epidermal and hair follicle keratinocytes but not in other cells after skin injury⁵. Moreover, our immunohistochemical data showed that REG3A was primarily expressed in epidermal and hair follicle keratinocytes in skin wounds, but not in dermis (Figure 1e). Furthermore, accumulating evidence has shown that REG3A is highly expressed in epithelial cells, paneth cells in gut, enterocytes, β cells and cancer cell lines, but not in immune cells that were recruited into the dermis¹⁻⁷.

General points that need addressing/clarification:

Comment: Fig 1 A You cannot see the wound edge in these images and this is required to be able to tell that it is wound and not normal tissue.

Response: As the reviewer suggested, we redid all the histological analyses and showed the wound edges (right sides) in all sections in Figure 1b in the revised version.

Comment: What type of wound is this and how old is it?

Response: The wounds we showed here were acute wounds in normal and diabetic patients. The information of patients is shown in the following:

Patients	Normal patients with acute injury	Diabetic patients with acute injury
Gender and age	1. Male, 52-year-old 2. Female, 34-year-old 3. Male, 31-year-old	1. Male, 61-year-old 2. Male, 61-year-old 3. Male, 51-year-old

We also added this information in Supplementary information.

Comment Fig 1 I - are these images the median values from the data set or the best/worst? How many mice were used in the wound healing studies - it is not clear.

Response: These images are the median values from the data set. 7 mice for Nor and DM groups while 6 mice for Nor+R and DM+R groups were tested. We have added this information in Figure legends.

Comment Sup 1. b, c & d wound edge images - we really need to be able to see the wound edge and we can't. This could be anywhere.

Why have the different mag between b & c?

Response: As the reviewer suggested, we showed the wound edges of Supplementary Fig.1b (now Figure 1e in the revised version), deleted Supplementary Fig.1c (but show wound edges in Figure 1 attached to this point-by-point response). The samples in Supplementary Fig.1d were the same as in Figure 1e. To see leukocytes infiltration in the dermis we did not show the wound edges but showed them in the same magnification as in Figure 1e.

Comment b seems to show some staining in the dermis - what structures is this in? Blood vessels? It is not clear from the pictures but this could also be interesting and is not addressed.

Response: Figure S1b showed the epidermis of the skin. We redid the immunohistochemical staining of these sections and relocated this figure in Figure 1e.

Comment Fig 2 a The legend describes this as day 3 wound but the figure appears to be a time course, 0, 1 3, 5 & 7.

I find this result rather odd that REG3 turns on upon wounding and then remains constant for 7 days in normal wounds. I would have expected changes in line with increasing and then decreasing as inflammation resolves.

Response: We apologize for the mistake and corrected the information in the figure legend.

We re-evaluated the production of RegIII γ and IL-33 in skin wounds by ELISA and showed that the production of IL-33 was significantly increased in a time-dependent manner with the maximum induction observed at day 1 but with slight

decrease at day 3 (Figure 2c). In line with this, the production of RegIII γ was significantly increased in a time-dependent manner with the maximum induction observed at day 5 but decreased at day 7 (Figure 2c) although it was hard to observe in Supplementary Figure 2a. All these data suggest that skin injury increases IL-33 production in keratinocytes, and IL-33, in turn, acts on keratinocytes to induce RegIII γ expression.

Comment Fig 2b is supposed to be a skin wound but I can see no details of the tissue to tell what cell types these are or where they are in relation to the wound.

Response: We have removed the data of db/db mice from the main Figures. We redid the immunohistochemical staining of IL-33 production in STZ-induced type 1 diabetic mice and showed that IL-33 production was markedly increased in the epidermal keratinocytes at wound edges from normal mice, but markedly decreased in the epidermal keratinocytes at wound edges from STZ-induced type 1 diabetic mice (Figure 2d)

Comment P 7 high glucose experiment. There are no experimental details given in the methods for this. How long were the cells cultured for as it can take some time (several days) for the glucose to take effect. This experiment could really do with a Mannitol control for osmolarity. Line 2 & 5 Fig 3k should be 2k. Fig 2 e is the scale bar correct at 25um or are the nuclei huge?

Response: As suggested by the reviewer, we added the mannitol as the control and re-tested the inhibitory effect of high glucose on the inhibition of IL-33 production by IL-17. Figure 2l and Supplementary Figure 3b showed that high glucose but not mannitol markedly inhibited IL-17-induced IL-33 expression in neonatal and adult human keratinocytes. We also added the experimental details in Methods section and corrected typos of Fig 2k.

For Figure 2e, the scale bar was corrected. Green staining showed REG3A expression, blue staining showed nuclei. To maintain clarity and brevity of the original report, we deleted Figure 2e.

Comment Fig 2 L - what type of diabetic mice are these? Needs to be clarified as 3 types are being used. Same goes for Fig 3 and Fig 4 I

Response: Except Supplementary Figure 1, STZ-induced type 1 diabetic mice were used in all other figures. We have clarified this in Figure 1 legend and in the main text.

Comment Fig 4 f. I am unable to see the wound edge in these images. I am unable to see anything in the db/db images.

Response: The samples used in Figure 4f were the same as samples used in Figure 1 attached to this point-by-point response. Since the reviewer thought that STZ-induced type 1 diabetic mice were sufficient, we deleted this data from db/db mice to streamline the message.

References

1. Gallo, R.L. & Hooper, L.V. Epithelial antimicrobial defence of the skin and intestine. *Nature reviews. Immunology* **12**, 503-516 (2012).
2. Vaishnava, S., Behrendt, C.L., Ismail, A.S., Eckmann, L. & Hooper, L.V. Paneth cells directly sense gut commensals and maintain homeostasis at the intestinal host-microbial interface. *Proceedings of the National Academy of Sciences of the United States of America* **105**, 20858-20863 (2008).
3. Cash, H.L., Whitham, C.V., Behrendt, C.L. & Hooper, L.V. Symbiotic bacteria direct expression of an intestinal bactericidal lectin. *Science* **313**, 1126-1130 (2006).
4. Choi, S.M., *et al.* Innate Stat3-mediated induction of the antimicrobial protein Reg3gamma is required for host defense against MRSA pneumonia. *The Journal of experimental medicine* **210**, 551-561 (2013).
5. Lai, Y., *et al.* The antimicrobial protein REG3A regulates keratinocyte proliferation and differentiation after skin injury. *Immunity* **37**, 74-84 (2012).
6. van Beelen Granlund, A., *et al.* REG gene expression in inflamed and healthy colon mucosa explored by in situ hybridisation. *Cell and tissue research* **352**, 639-646 (2013).
7. Hervieu, V., *et al.* HIP/PAP, a member of the reg family, is expressed in glucagon-producing enteropancreatic endocrine cells and tumors. *Human pathology* **37**, 1066-1075 (2006).
8. Suga, H., *et al.* TLR4, rather than TLR2, regulates wound healing through TGF-beta and CCL5 expression. *Journal of dermatological science* **73**, 117-124 (2014).
9. Lai, Y., *et al.* Commensal bacteria regulate Toll-like receptor 3-dependent inflammation after skin injury. *Nature medicine* **15**, 1377-1382 (2009).
10. Eming, S.A., Martin, P. & Tomic-Canic, M. Wound repair and regeneration: mechanisms, signaling, and translation. *Science translational medicine* **6**, 265sr266 (2014).
11. Caporali, A., *et al.* Soluble ST2 is regulated by p75 neurotrophin receptor and predicts mortality in diabetic patients with critical limb ischemia. *Arteriosclerosis, thrombosis, and vascular biology* **32**, e149-160 (2012).
12. Watanabe, N., *et al.* Prediction of gestational diabetes mellitus by soluble (pro)renin receptor during the first trimester. *The Journal of clinical endocrinology and metabolism* **98**, 2528-2535 (2013).

Figures for point-by-point responses:

Figure 1

[redacted]

Figure 2

[redacted]

Figure 3

[redacted]

Figure 4

[redacted]

Reviewer #1 (Remarks to the Author):

In the revised manuscript entitled "Hyperglycemia inhibits REG3A expression to exacerbate TLR3-mediated skin inflammation in diabetes", Wu et al., reported a novel role of the antimicrobial protein REG3A controlling TLR3-mediated inflammation after skin injury. The authors have addressed satisfactorily all questions raised by this reviewer, and I have no further concerning.

Reviewer #2 (Remarks to the Author):

The authors have addressed most of the concerns of the initial review, including performing new experiments to repeat and confirm results. However, a major concern is that the results rely heavily on western blot analyses without appropriate presentation of densitometry data and statistical analyses for important comparisons, which was not adequately addressed in the revised manuscript. In addition, there was one instance in which the interpretation of results did not match the data and a couple of minor textual errors in the figures. These are described below.

Major Revisions

1) For almost all of the western blots, the authors provided single densitometry numbers based on image J analysis of the bands compared with beta-actin. This is not how western blot densitometry data is typically presented and it does not allow for appropriate statistical comparisons for important results. Please include bar graphs with error bars and statistical analyses for the densitometry data in Figs. 2i, 2j, 2k, 2l, 2n, 5d, 5e, 5f, 5g, 5h, 6c, 6i, 6j, 6o, 7a, 7b, 7e, 7f, Suppl. Fig. 2b, Fig. 3b, 3c and Fig. 5h. Please remove all other single densitometry numbers from the remainder of the western blots as they are not necessary.

Minor Revisions

1) Fig. 7g - The text describes a significant difference between DM+R and DM+SSG+R treatment groups but the graph does not support this statement (no significant difference is found between these groups). Please modify this statement in the text to support the proper interpretation of the data or repeat the experiment to achieve statistical significance between DM+R and DM+SSG+R treatment groups

2) Suppl Fig 2c. There is a misplaced "gamma" symbol overlying the text for "REG3A"

3) Suppl Fig. 3b - Is the treatment in the three right lanes supposed to be labeled rhIL-17? Please fix this typo.

Reviewer #3 (Remarks to the Author):

On the whole the authors have made a big effort to address my comments and questions. The revised manuscript is now easier to read and is less confusing. However, not all of my points around the histology have been fully addressed.

Comment: The presentation of histological data and staining is problematic. Whilst all of the figure legends describe wound edge, a wound edge is never shown! You really need to be able to see the wound edge and changes that take place are at the wound edge and you need to be able to see them declining away from the wound edge. All of the images that are presented could be anywhere in intact skin. Also it is important to consider what is happening in the dermis as this plays a very important role (probably greater than the epidermis) in inflammation and wound healing. Why has this been ignored in this study? As it stands the way the data has been presented I am unable to properly review the results of the histology.

Response: Overall we believe these critiques have now been resolved as we redid all the histological analyses and showed the wound edges in all sections (Figure 1b,e and Figure 2d,e)

I still have problems with the presentation of the histology in figures 1 & 2. The images are far too small for me to see the required detail even when I blow them up on my screen. In figure 2 D&E you can see clear convincing nuclear staining but in figure 1 B&E all I can see is a murky brown with a lot of background staining from the HRP rather than the contrast of believable staining we see in Fig 2.

Again we are not told how old these wounds are.

My additional concern is that the staining seems to be fairly uniform all along the epidermis and not particularly related to the wound edge as one might expect in a wound healing response.

Taken together the histology part of the story is not very convincing.

Comment: What type of wound is this and how old is it?

Response: The wounds we showed here were acute wounds in normal and diabetic patients. The information of patients is shown in the following:

This response does not tell me how old the wounds are, which is what I was asking.

Comment Sup 1. b, c & d wound edge images - we really need to be able to see the wound edge and we can't. This could be anywhere.

Why have the different mag between b & c?

Response: As the reviewer suggested, we showed the wound edges of Supplementary Fig.1b (now Figure 1e in the revised version), deleted Supplementary Fig.1c (but show wound edges in Figure 1 attached to this point-by-point response). The samples in Supplementary Fig.1d were the same as in Figure 1e. To see leukocytes infiltration in the dermis we did not show the wound edges but showed them in the same magnification as in Figure 1e.

This response partly addresses my question by saying it is 2mm from the wound edge but I was asking to see the wound edge so I could judge the effect on the inflammatory response. The counts of leukocytes in S1 of 2,000-4,000 per mm² seem way out of order. From the images this is hard to believe. Also we are not told how old these wounds are or how many mice N there were - just that it is representative of two independent experiments.

Minor points:

The text and figure legend around Fig4J are quite confusing and could be written more clearly.

Methods - was the hair really plucked from the mice?

Point-by-Point Response

Reviewer #2:

The authors have addressed most of the concerns of the initial review, including performing new experiments to repeat and confirm results. However, a major concern is that the results rely heavily on western blot analyses without appropriate presentation of densitometry data and statistical analyses for important comparisons, which was not adequately addressed in the revised manuscript. In addition, there was one instance in which the interpretation of results did not match the data and a couple of minor textual errors in the figures. These are described below.

Major Revisions

Comment 1) For almost all of the western blots, the authors provided single densitometry numbers based on image J analysis of the bands compared with beta-actin. This is not how western blot densitometry data is typically presented and it does not allow for appropriate statistical comparisons for important results. Please include bar graphs with error bars and statistical analyses for the densitometry data in Figs. 2i, 2j, 2k, 2l, 2n, 5d, 5e, 5f, 5g, 5h, 6c, 6i, 6j, 6o, 7a, 7b, 7e, 7f, Suppl. Fig. 2b, Fig. 3b, 3c and Fig. 5h. Please remove all other single densitometry numbers from the remainder of the western blots as they are not necessary.

Response: As the reviewer suggested, we have removed all densitometry numbers from western blots and shown bar graphs with error bars and statistical analyses for all blots' densitometry in Supplementary Information.

Minor Revisions

Comment 1) Fig. 7g - The text describes a significant difference between DM+R and DM+SSG+R treatment groups but the graph does not support this statement (nosignificant difference is found between these groups). Please modify this statement in the text to support the proper interpretation of the data or repeat the experiment to achieve statistical significance between DM+R and DM+SSG+R treatment groups.

Response: We combined the data of two experiments and showed the statistical significance between DM+R and DM+SSG+R treatment groups (9 mice for each group).

Comment 2) Suppl Fig 2c. There is a misplaced "gamma" symbol overlying the text for "REG3A"

Response: Thank you for the note. "gamma" is for "rhIL-36 γ ". We have corrected it.

Comment 3) Suppl Fig. 3b - Is the treatment in the three right lanes supposed to be labeled rhIL-17? Please fix this typo.

Response: Yes, the treatment in three right lanes was with rhIL-17. We have fixed it.

Reviewer #3:

On the whole the authors have made a big effort to address my comments and questions. The revised manuscript is now easier to read and is less confusing. However, not all of my points around the histology have been fully addressed.

***Comment:** The presentation of histological data and staining is problematic. Whilst all of the figure legends describe wound edge, a wound edge is never shown! You really need to be able to see the wound edge and changes that take place are at the wound edge and you need to be able to see them declining away from the wound edge. All of the images that are presented could be anywhere in intact skin. Also it is important to consider what is happening in the dermis as this plays a very important role (probably greater than the epidermis) in inflammation and wound healing. Why has this been ignored in this study? As it stands the way the data has been presented I am unable to properly review the results of the histology.*

***Response:** Overall we believe these critiques have now been resolved as we redid all the histological analyses and showed the wound edges in all sections (Figure 1b,e and Figure 2d,e)*

I still have problems with the presentation of the histology in figures 1 & 2. The images are far too small for me to see the required detail even when I blow them up on my screen. In figure 2 D&E you can see clear convincing nuclear staining but in figure 1 B&E all I can see is a murky brown with a lot of background staining from the HRP rather than the contrast of believable staining we see in Fig 2.

Response: We redid the immunohistochemical analysis of these samples and showed the new histology data with clear staining of keratinocytes and epidermis in Figure 1b&e. Due to the space limitation, the images in Figure 1b&e were still present in a combination of 400X and 100X magnifications, but the images with 400X magnification and the images with 100X magnification were present separately in Figure 1 attached to this point-by-point response.

The reason why the reviewer can see clear nuclear staining in Figure 2d&e but not in Figure 1b&e is that the samples in Figure 1b&e were stained by REG3A or RegIII γ while the samples in Figure 2d&e were stained by IL-33. REG3A or RegIII γ was primarily expressed in the cytoplasm of keratinocytes while IL-33 was primarily expressed in the nucleus of keratinocytes.

***Comment:** Again we are not told how old these wounds are.*

Response: All mouse samples were from day-3 skin wounds. For human samples, the samples from normal patients were taken 1(two patients) or 11 days after acute injury while the samples from diabetic patients were taken 5 or 9 or 14 days after acute injury. We have added the information in Figure legends.

***Comment:** My additional concern is that the staining seems to be fairly uniform all along the epidermis and not particularly related to the wound edge as one might expect in a wound healing response.*

Taken together the histology part of the story is not very convincing.

Response: We did the immunohistochemical analysis of skin wounds from 20 normal and diabetic mice and all were the same as we showed in Fig 1e. One explanation will be that RegIII γ is a secretory protein and can be secreted out of keratinocytes, that's why it looks fairly uniform all along the epidermis around wound edges. Alternatively, RegIII γ was induced by pro-inflammatory cytokine IL-33. Therefore, increased RegIII γ should be observed in the area where IL-33 was increased. Since IL-33 was fairly uniform all along the epidermis of skin wounds (Fig. 2d), the expression of RegIII γ would also show the similar expression pattern in skin wounds. Thirdly, the samples we shown in Fig. 1b&e and Fig. 2d&e were about 1.2mm thick skin around wound edges. Although how far inflammatory responses could reach in skin wounds remains unclear, at least inflammatory responses can be observed within 1.2mm skin far from wound edges (See Figure 2 attached to this point-by-point response).

Comment: What type of wound is this and how old is it?

Response: The wounds we showed here were acute wounds in normal and diabetic patients.

The information of patients is shown in the following:

This response does not tell me how old the wounds are, which is what I was asking.

Response: Skin wounds from normal patients were taken 1 (two patients) or 11 days after acute injury while skin wounds from diabetic patients were taken 5 or 9 or 14 days after acute injury. We have added the information in Figure legends.

Comment Sup 1. b, c & d wound edge images - we really need to be able to see the wound edge and we can't. This could be anywhere.

Why have the different mag between b & c?

Response: As the reviewer suggested, we showed the wound edges of Supplementary Fig.1b (now Figure 1e in the revised version), deleted Supplementary Fig.1c (but show wound edges in Figure 1 attached to this point-by-point response). The samples in Supplementary Fig.1d were the same as in Figure 1e. To see leukocytes infiltration in the dermis we did not show the wound edges but showed them in the same magnification as in Figure 1e.

This response partly addresses my question by saying it is 2mm from the wound edge but I was asking to see the wound edge so I could judge the effect on the inflammatory response. The counts of leukocytes in S1 of 2,000-4,000 per mm² seem way out of order. From the images this is hard to believe. Also we are not told how old these wounds are or how many mice N there were - just that it is representative of two independent experiments.

Response: It is known that the inflammatory responses in skin wounds do not only occur at wound edges, and we also observed that the inflammatory responses happened within 1-2mm skin far from wound edges, especially in skin wounds of diabetes (See Figure 2 attached to this point-by-point response). Moreover, in our study we tried to show that REG3A or RegIII γ inhibited TLR3-dependent inflammation in skin wounds not only at wound edges. Figure 1b&e showed that REG3A or RegIII γ was expressed in keratinocytes of skin wounds including wound edges, and ELISA or RT-PCR data in Fig. 1f&g and Fig.3d clearly showed that RegIII γ significantly inhibited TLR3-mediated inflammatory response in skin wounds of nondiabetic and diabetic mice. So far, we have the slides that can clearly

show the inflammatory responses in skin wounds including wound edges of nondiabetic and diabetic mice (See Figure 2 attached to this point-by-point response), but we do not have slides that can clearly show both wound edges and wounds from nondiabetic and diabetic mice treated with RegIII-gamma even though the images in Supplementary Fig.1d we chose were close to wound edges. Since the inhibitory effect of REG3A/ RegIII γ did not only occur in wound edges, we don't think it is necessary to take 1-2 months for repeating the experiment of counting leukocytes in wound edges of nondiabetic and diabetic mice treated with RegIII γ .

The number of leukocytes in Supplementary Fig. 1d of 2,000-4,000 per mm² was correct. The scale bar of inset in Fig. S1d was 50 μ m. There were 5-10 leukocytes in 2500 (50x50) μ m². Therefore, it would be 2,000 (5X400)-4,000 (10X400) leukocytes per mm² (1mm=1000 μ m, 1000X1000 /50x50=400).

The wounds were day-3 wounds and 4 mice for each group. We have added the information in Supplementary Figure 1 legend.

Minor points:

Comment: The text and figure legend around Fig4J are quite confusing and could be written more clearly.

Response: As the reviewer suggested, we have rewritten the text and figure legend of Fig 4J to make it more clearly.

Comment: Methods - was the hair really plucked from the mice?

Response: Sorry for the wrong description. Mouse fur was removed by using chemical depilation. We have corrected it.

Figures for point-by-point responses:

Figure 1 [redacted]

Figure 2 [redacted]

Reviewer 3

I remain concerned about the wound histology material. The 3 day mouse wounds are fine for comparison but the human wounds are not. A 1 day and 11 day wound is very different and I would have expected the 11 day wound to be completely healed from normal skin but the images do not give that appearance. Similarly these wounds can not reasonably be compared to the 5, 9 and 14 day wounds, which should also be healing but they do not have that appearance.

It is odd that the 3 day mouse wound edge tissue is hyperthickened in both normal and diabetic mice. You would normally expect to see a thin tongue of migratory cells but we do not see that. Whilst I might have expected this in the diabetic mice I would not expect it in normal mice.

I still believe that the inflammatory response at the wound edge is more important than 2mm away. For me that would be additional information on how far inflammation had spread.

Point-by-point response

Comment 1: I remain concerned about the wound histology material. The 3 day mouse wounds are fine for comparison but the human wounds are not. A 1 day and 11 day wound is very different and I would have expected the 11 day wound to be completely healed from normal skin but the images do not give that appearance. Similarly these wounds can not reasonably be compared to the 5, 9 and 14 day wounds, which should also be healing but they do not have that appearance.

Response: Although all human samples used for this study were from acute wounds, the ‘normal’ control samples were excised from non-diabetic following a minor acute injury from an auto accident. The diabetic samples were excised from diabetic patients following a minor acute traumatic injury. Controls had undergone surgery related to the auto accident within 1 or two days after they were admitted to the hospital. We thereby excised most of ‘normal’ samples within 1 or 2 days after initial injury related to the accident. For diabetic patients, they usually were not admitted to the hospital after minor traumatic injuries, so we only excised the samples at 5 or 9 or 14 days after injury. Due to these uncontrollable factors, it is difficult to obtain samples within a more defined time range.

We agree that 11-day wound from a ‘normal’ control patient should be completely healed. The day-11 control wound was excised from the patient during his second surgery because the wound of this patient did not heal after the first surgery. This suggests that this patient has some problem with skin wound healing, or secondary complications from the accident. Therefore, we obtained another ‘normal’ control sample from a 2-day skin wound from Changhai Hospital and replaced the day-11 sample in Figure 1b.

For diabetic patients, it has been established that cutaneous wound healing is usually delayed. The current study is designed to elucidate a potential mechanism by which wound healing is delayed in diabetic patients, and demonstrates that defective REG3A amplifies inflammation to impair wound healing under diabetic conditions. Therefore, we believe it is reasonable that we do not observe complete healing in day 5, 9 and 14 day skin wounds.

Comment 2: It is odd that the 3 day mouse wound edge tissue is hyperthickened in both normal and diabetic mice. You would normally expect to see a thin tongue of migratory cells but we do not see that. Whilst I might have expected this in the diabetic mice I would not expect it in normal

mice.

Response: In general, the epidermis of diabetic mice in day-3 acute wounds is thinner than the epidermis of normal control mice in day-3 acute wounds, but these morphological differences observed may be due to individual differences between mice. We have also observed this phenomenon in our studies (see Figure 1 attached to this-point-by-point responses). We thereby changed the second sample in diabetic group in Figure 1e, which more clearly represented the morphology under diabetic condition. Regarding the observation of a thin tongue of migratory cells, it is hard to observe this in day-3 wounds, but it can be more readily observed in day-5 wounds from normal control mice (see Figure 1 attached to this-point-by-point responses).

Comment 3: I still believe that the inflammatory response at the wound edge is more important than 2mm away. For me that would be additional information on how far inflammation had spread.

Response: We agree that the inflammatory response at the wound edge is important, but that information regarding the inflammation distal from the primary wound is also critical. From H&E staining of day-5 skin wounds, we observed leukocyte infiltration within 2.56mm of the wound edge (see Figure 2 attached to this-point-by-point responses).

Figure for point-by-point responses:

Figure 1 [redacted]

Figure 2 [redacted]